# How to Improve Physico-Chemical Properties of Silk Fibroin Materials for Biomedical Applications?—Blending and Cross-Linking of Silk Fibroin—A Review

**DOI:** 10.3390/ma14061510

**Published:** 2021-03-19

**Authors:** Sylwia Grabska-Zielińska, Alina Sionkowska

**Affiliations:** 1Department of Physical Chemistry and Physicochemistry of Polymers, Faculty of Chemistry, Nicolaus Copernicus University in Toruń, 87-100 Toruń, Poland; 2Department of Chemistry of Biomaterials and Cosmetics, Faculty of Chemistry, Nicolaus Copernicus University in Toruń, 87-100 Toruń, Poland; as@chem.umk.pl

**Keywords:** silk fibroin, biopolymers, polymer blends, biomaterials, polymer composites

## Abstract

This review supplies a report on fresh advances in the field of silk fibroin (SF) biopolymer and its blends with biopolymers as new biomaterials. The review also includes a subsection about silk fibroin mixtures with synthetic polymers. Silk fibroin is commonly used to receive biomaterials. However, the materials based on pure polymer present low mechanical parameters, and high enzymatic degradation rate. These properties can be problematic for tissue engineering applications. An increased interest in two- and three-component mixtures and chemically cross-linked materials has been observed due to their improved physico-chemical properties. These materials can be attractive and desirable for both academic, and, industrial attention because they expose improvements in properties required in the biomedical field. The structure, forms, methods of preparation, and some physico-chemical properties of silk fibroin are discussed in this review. Detailed examples are also given from scientific reports and practical experiments. The most common biopolymers: collagen (Coll), chitosan (CTS), alginate (AL), and hyaluronic acid (HA) are discussed as components of silk fibroin-based mixtures. Examples of binary and ternary mixtures, composites with the addition of magnetic particles, hydroxyapatite or titanium dioxide are also included and given. Additionally, the advantages and disadvantages of chemical, physical, and enzymatic cross-linking were demonstrated.

## 1. Introduction

Biopolymeric materials are recurrently used in the biomedical field, e.g., for wound healing, gene therapy, drug delivery, bone tissue engineering, cartilage, nerve, and eye regeneration [1,2,3,4,5,6]. Increasing interest in new materials with potential use in the biomedical field has been observed during the last few years [1,2,3,4,5,7]. In this manuscript, we reviewed different usages of silk fibroin-based materials blended with other biopolymers and cross-linked with chemical agents, in biomedical applications based on previous research. However, one area of biomedical applications deserves a greater introduction—bone tissue engineering.

Most people in the world suffer from diseases related to muscles, joints, and bones. Demographic studies indicate aging of the population. As a result of these predictions, it is estimated that the number of people with joint and bone diseases will increase [8]. The skeleton is the support of the whole human body, and the place where the muscles are attached, it plays an important role in the movement process, transferring the strength of the muscles and allowing for movement. It also protects the vital organs of the human body from damage [9]. The bone system carries many important functions in the human body, therefore, material engineering, including tissue engineering and reconstructive medicine, is developing at an increasing rate. Treatment of bone defects resulting from diseases, injuries, or infections is a big challenge for modern science. All the time, we are searching for the best materials for the production of scaffolds that support the regeneration of damaged tissue. Tissue engineering is science, the aim of which is the use of medical and material engineering knowledge to obtain functional substitutes for damaged human tissue [10,11]. In response to the needs of tissue engineering, the aim is to try to obtain new materials with features needed to create the ideal material (in terms of bioactivity, bioresorbability, biocompatibility, degradability to non-toxic degradation products). Nowadays bone-cartilage implants are produced from different types of material: ceramic, metallic, and polymeric [12,13,14]. Materials based on polymers are used to obtain implants for small bone or cartilage tissue cavities. Such materials are called scaffolds [15]. Three-dimensional artificial scaffolds supporting the regeneration of damaged tissue should fulfil the following functions:✓ formation and movement of cells,✓ having an influence on development and differentiation of cells,✓ merging the cells in the tissue,✓ mechanical support for proteins and cells.

Scaffolds should be characterized by a connection of mechanical, chemical, and biological properties [16,17,18]. One such feature is the porosity, whereby cells can penetrate deep into the material. This leads to cell proliferation and the reconstruction of tissues [19]. Increasingly, natural polymers are used for this type of biomaterial because of their biocompatibility. Biomaterials based on biopolymers can be biodegradable due to several chemical moieties in their structure. The implant slowly degrades while reconstituting the tissue. When the biomaterial is biodegradable, degradation products are non-toxic and biomaterial does not lead to immune-response, we can eliminate the need to remove the implant after tissue reconstruction [7]. The ability of cells to penetrate the entire volume of the implant (adhesion to the material) is determined by bioactivity of the natural polymers [20].

Macromolecules that occur in living organisms (or are produced by living organisms) are called natural polymers [7]. A number of protein polymers (silk fibroin [21,22,23,24,25], collagen [7,26,27,28,29], elastin [29], gelatin [30]) and polysaccharides (chitosan [31,32], chondroitin sulphate [29], sodium alginate [31], hyaluronic acid [29,31,32]) are used in biomaterial production. Polymers, derived from natural sources, are characterized by properties desired in tissue engineering: biocompatibility, biodegradability, and lack of immune responses after introduction into the human body [32,33]. In the last few years, silk fibroin-based biomaterials have attracted increasing attention. By compiling the statistics of the articles searched from the database of the Web of Science using the keywords of “silk fibroin” and “tissue regeneration” it was found that the annual number of articles was less than 10 before 2006. The number of articles increased continually after 2006 and reached 155 in 2020 (Figure 1). This dependence indicates that silk fibroin-based biomaterials are becoming more popular in basic research of tissue engineering.

In this review, a brief introduction of silk fibroin and different physical forms of silk fibroin-based materials are given. Then, mixing with other polymers and cross-linking of silk fibroin-based materials, are summarized. Ultimately, an outlook of the directions of the development of silk fibroin-based biomaterials for future applications is provided. It is foreseen that this review can provide an exhaustive compendium regarding the main aspects of silk fibroin-related research and, therefore, stimulate its future growth and applications.

## 2. Silk Fibroin

Silk fibroin (SF) is a typical, natural polymer belonging to the group of proteins. It shows unique chemical and physical properties [14,34]. It has excellent mechanical properties. Silk fibroin is built of repetitive protein sequences, which result in providing structural roles in arachnid and crustacean life: web formation, egg protection, cocoon formation, building nest, and traps [34].

Silk fibroin was used as sutures in the biomedical field for decades [14]. Silk fibroin-based materials are attractive and widely used for bone tissue regeneration due to properties such as slow degradability, remarkable mechanical strength, and thermal stability, versatility in processing, easy modification [14,34]. They also show genetically tailorable composition, sequence, and permeability to oxygen and water [14]. These are the advantages of silk fibroin. Nevertheless, silk fibroin has some disadvantages, like necessity of purification, because residual of sericin in the material can provide contamination and problems with biocompatibility [14].

## 3. Silk Fibroin Sources

The two most common sources to obtain silk fibroin are: *Bombyx mori* cocoons and *Nephila clavipes* webs. *Bombyx mori* (the domestic silk moth) is an insect from the moth family *Bombycidae*. The silkworm is the larva or caterpillar of a silk moth [35]. *Nephila clavipes* is generally famed as the golden silk orb-weaver or banana spider. These spiders are excellent web-builders, producing and utilizing seven different types of silk [36].

### 3.1. Cocoon (Bombyx Mori) Silk

Mulberry silkworm (*Bombyx mori*) is the major producer of silk throughout the world [37]. Thanks to the well-established sericulture process, about 400,000 tons of dry silkworm *Bombyx mori* cocoons are available worldwide per annum for the textile industry and thus for applications in the biomedical field [34]. The raw silk fiber forming a *Bombyx mori* cocoon is composed of fibroin (75–83%) and sericin (17–25%). Fibroin is a core of two fibers with 10 to 25 µm in diameter, coated with sericin, acting as glue, that holds fibroin fibers together. Silk fibroin fibers consist of two proteins which are present in a 1:1 ratio and linked by a single disulfide bond (light and heavy chain: ~26 kDa and ~390 kDa, respectively) [14,34,37]. Sericin is capable of causing allergic and immunologic reactions in humans [34,37]. Sericin has to be removed by the de-gumming process through boiling in water, soap, or acid treatments [37,38]. Silk fibroin is mainly made of repeatable sequences of amino acids, such as serine (12%), alanine (30%), and glycine (43%) [39,40]. It can arrange itself into two structural conformations: silk I (α-helix) and silk II (β-sheets) [40]. Silk I can be soluble in water, but silk II can be dissolved using only aqueous inorganic salt (LiBr, LiSCN) or concentrated acids. It is because those solutions have the ability to break hydrogen bonds stabilizing a β-sheets silk fibroin structure [41,42,43]. The preparation process of silk fibroin from *Bombyx mori* cocoons is shown in Figure 2. The procedure is similar as in Q. Lu, et al. [37,44] with small modifications. First, the cocoons have to be boiled in a Na_2_CO_3_ aqueous solution (60 min; 100 °C; two times). Then, the cocoons have to be boiled in an alkaline soap solution (30 min; 100 °C), washed in tap and distilled water, and boiled in distilled water (20 min; 100 °C). This procedure must be repeated three times. Subsequently, the degummed silk is dried at room temperature and humidity for 48 h. Regenerated silk fibroin can be dissolved in an appropriate solvent system, such as 9.3 M lithium bromide, lithium thiocyanate aqueous solution, Ajisawa’s reagent (CaCl_2_:H_2_O:C_2_H_5_OH; 1:8:2.5 molar ratio), N-methyl morpholine-N oxide, calcium nitrate aqueous solution, hexafluoroisopropanol (HFIP), and ionic liquids. The time and temperature of this process depend on the solvent and solubility power of the solvent [23]. After that, dissolved silk fibroin solution can be dialyzed against pure, distilled water, and the aqueous solution of silk fibroin is obtained [37].

### 3.2. Spider (Nephila Clavipes) Silk

Silk derived from *Nephila clavipes* has been studied in detail. It does not have a sericin coating, is safer than *Bombyx mori* silk, and elicits almost no immunological response [45]. It can be processed via formation of a spidroin solution or used in natural fiber form [46]. Spider silk is characterized by remarkable thermal stability and excellent mechanical properties, especially high tensile strength [37,45]. In *Nephile clavipes* silk, two distinct proteins with similar sequences can be found: MaSp1 (major ampullate spidroin 1) and MaSp2 (major ampullate spidroin 2). They have different repeat units and possibly distinct mechanical functions. Glycine (G) (above 42%) and alanine (A) (above 25%) are the main amino acids in the spidroin core. Glutamine (Q), serine (S), leucine (L), valine (V) proline (PR), and tyrosine (Y) are the remaining amino acids found in the core. The β-sheet structure is mainly formed by alanine. MaSp1 consists GGX motif (where X = Y, L, or Q), arranged in flexible 310-helical structures. MpSp2 have a GPGXX motif (where X = G, Q, L, or Y), which consists repeatedly in sequence, and they arrange elastic β-turn spiral or helical structures [47,48,49].

## 4. Types of Silk Fibroin Based Biomaterials

Silk fibroin is used in many fields of biomedicine. It is exploited in bone tissue engineering, eye regeneration, nerve regeneration, skin tissue engineering, cartilage regeneration, vascular tissue engineering, spinal cord tissue engineering, gene therapy, and biological drug delivery [23,50,51] (Figure 3).

Silk fibroin can be processed into different material morphologies: sponges, gels, membranes, powders, and microspheres [34]. The fibers must first be dissolved in aqueous systems, followed by reprocessing into desired material formats [34]. In this part of article, the most common material morphologies will be described. The most relevant types of morphologies with method processing and applications will be described in Table 1.

### 4.1. Fibres

Reeling from cocoons is the method to obtain silk fibroin fibers [34]. These fibers are characterized as smooth, light, breathable, moisturizing, biocompatible, and degradable. They are widely used in the textile industry, handicrafts, and so on [52]. However, these type of silk fibroin fibers have some disadvantages—they are susceptible to aging and mildew. A better way to obtain silk fibroin fiber is to prepare resolubilized silkworm cocoon silk or to use genetically engineered variants of silk [53], which can be spun into fibers. They are characterized by poor mechanical properties and weak cell attachment. Pure silk fibroin promotes microbial growth. There is a way to improve the properties of pure silk fibroin fibers—effective modification and treatment have to be applied. Figure 4A shows the morphology of silk fibroin fiber obtained by a spinning method with formic acid as solvent [54]. Electrospinning can be used to generate silk from the silkworm, spider, and genetically engineered silks. The development in the scientific and industrial world has led to the possibility of obtaining nanofibers (Figure 4B). They are attractive candidates for biomedical applications, including tissue engineering, wound healing, and drug delivery systems [52,53]. They are characterized by a new set of properties, e.g., excellent mechanical and thermal properties, high specific surface area, increased surface energy, and enhanced electrical conductivity [52,53]. Solubilized fibroin can also be transformed into films, mats, gels, membranes, and porous scaffolds [52,53].

### 4.2. 3D Porous Structures

Silk fibroin three-dimensional porous structures, namely scaffolds, sponges, or foams, are attractive for biomedical applications, especially for tissue engineering, because of their mechanical and thermal properties, biocompatibility, and degradability [55]. They are important for nutrient and waste transport, cell attachment, migration, and proliferation. Silk fibroin 3D scaffolds can be prepared from fibroin solution by lyophilization method, particulate-leaching, gas foaming, or electrospinning [24,55,56]. They can also be obtained by 3D printing. Due to these methods, sponges are characterized by large interior surface areas and interconnected pore spaces [24]. During the lyophilization process, pore size can be controlled by adjusting the pH of the solution, the freezing temperature and the amount of organic solvents [41] (Figure 5). Particulate-leaching is a simple method where sucrose or salt particles of a specific diameter, as porogens, are added to a polymer solution [56]. This technique and gas foaming method, give better control over pore structure than freeze-drying [51]. Electrospinning is a very popular method to obtain nanofibers. It is also a common technique to prepare tissue engineering scaffolds because it could provide a nanosize structure, which is desirable for cell migration and proliferation [57]. 3D printing can be used to fabricate the most ideal porous scaffolds with high porosity and good pore quantity and morphology. It is a computer-controlled layer-by-layer fabrication technique [57]. 3D porous silk fibroin-based scaffolds have been used in cartilage and bone tissue regeneration, due to success in achieving good control over pore size and porosity [53].

### 4.3. Particles

Silk fibroin is an ideal candidate for drug delivery systems because of its biocompatibility and controlled degradation. Microspheres, nanolayers, nanocoatings, nanoparticles, and microcapsules have been studied and reported as materials used in the biomedical field, especially in the controlled release of drugs [58]. Biodegradable silk microspheres are widely proposed for use in controlled-release drug delivery systems. Microspheres can be obtained by spray drying, water in oil (W/O) emulsion solvent evaporation, or water in oil emulsion solvent diffusion and ball-milling methods [59]. Silk micro and nanoparticles can be processed directly from fibers without using any chemicals [41] by the milling silk fiber method. They are being used in scaffolds for reinforcing and improving mechanical and biological properties [41]. Silk micro and nanoparticles can be produced by a lot of various methods, e.g., freeze-drying and grinding [60], spray drying [61], self-assembly [62], freeze-thawing [63], jet breaking [64], desolvation [65] or PVA blending (Figure 6) [66]. Regenerated silk particles are widely used in drug delivery systems [41]. Drug delivery systems via silkworm silk-based nanoparticles, with a size smaller than 100 nm, have been reported. They were used in anticancer therapy and they showed stability and non-toxicity [55]. There is a big challenge to connect two functions of silk fibroin particles. Work is underway to create particles that can simultaneously improve the functional properties of scaffolds and be drug carriers [41]. The biggest task is to prepare very fine and uniform silk particles, which is difficult in the milling method. The properties and application possibility of silk fibroin particles are dependent on the preparation method. The modification of method conditions results in different particle profiles. They can have various size, shape, charge, and lipophilicity [67].

### 4.4. Films

Silk fibroin-based films can be produced by the casting method [34,41,53,64]. In this process, *Bombyx mori* silk fibroin is widely used. It has to be degummed, washed with distilled water, dried, and dissolved in a suitable solvent. As a solvent, 9.3M LiBr [69,70], 10M LiSCN [71], Ca(NO_3_)_2_ x 4H_2_O [72], CaCl_2_/H_2_O/C_2_H_5_OH (molar ratio: 1:8:2.5) [37,73] solutions can be used. After that, dialysis against distilled water has to be prepared and an aqueous solution has to be poured onto glass or polystyrene Petri dishes. The film is obtained after the evaporation of the solvent. Figure 7 shows silk fibroin-based film (A) and a topographic image of it with a roughness of 5.1 ± 0.3 nm (B). CaCl_2_/H_2_O/C_2_H_5_OH mixture was used to dissolve dry silk fibroin. A rough surface is desirable in biomedical applications of films—it is advantageous for cell attachment [34]. To modify properties, such as mechanical properties and degradability and solubility of silk fibroin films, treatment with 50% methanol can be used [74]. This is the Langmuir-Blodgett (LB) process. The disadvantage of this treatment is the fact that materials embrittle with time [53]. To prepare more flexible materials, methanol treatment was avoided and an all-water annealing process for the films was utilized [75]. However, there are other silk fibroin film obtaining techniques, such as spin coating [76], vertical deposition [77], manual or spin assisted layer by layer assembly [78], and centrifugal casting [79].

### 4.5. Hydrogels

Hydrogels as three-dimensional polymer networks, which are able to swell in aqueous solutions but do not dissolve in these solutions, can be used in the biomedical field [41,53,65]. Silk fibroin hydrogels have been reported as materials for the delivery of cells and cytokines, drug release, cell encapsulation, fabrication of prosthesis for soft tissues, matrices for repairing and regenerating tissues and organs [41,55,65]. It was reported that silk fibroin hydrogels have suitable mechanical properties to prepare scaffolds for load bearing tissue engineering, such as cartilage regeneration [80]. They can be obtained by the sol-gel transition of aqueous silk fibroin solution in the presence of acids, ions, or other additives, by lyophilization and sonication techniques [41,81]. Some parameters during silk fibroin hydrogels obtained by the sol-gel transition can be controlled to regulate physico-chemical parameters of the final product. If protein concentration, temperature, pH, and amount of Ca^2+^ are higher, the rate of gelation is higher. However, the overall rate of gelation has to be controlled by osmotic stress [82]. The mechanical properties, morphology, and structure of silk fibroin hydrogel are dependent on the extension and rate of water removal [82]. Types of silk and specific salts are also relevant to gelation, therefore, calcium plays a role with silkworm silk and potassium with spider dragline silk [53].

### 4.6. Aerogels

Aerogels are ultralight materials comprised of a microporous solid in which the dispersed phase is a gas [83]. They are characterized by fine, open-pore structure resulting in low densities (0.003–0.15 kg/m^3^), large surface areas (500–1000 m^2^/g) and high porosity [84]. The surface area, pore size, mechanical and physico-chemical properties of aerogels can be tailored [84]. A few drying technologies for aerogel production can be outlined— ambient drying (ambient pressure, room or slightly elevated temperature), freeze drying (vacuum with *p* < 100 mbar; −70 < T<−20 °C), and direct supercritical drying (high temperature), supercritical drying by CO_2_ extraction (T > 31 °C, *p* > 74 bar) [84,85,86,87,88]. Aerogels can find application in many fields, including packaging materials, cosmetics and medicine in particular drug delivery systems, tissue engineering, and implants. There are only a few literature reports on silk fibroin-based aerogels. Li et al. [87] and Baldino et al. [89] reported silk fibroin-based aerogels and silk fibroin aerogels loaded with ascorbic acid, for nerve regeneration and nanomedicine applications, respectively. Najberg et al. [90] reported that aerogel sponges of silk fibroin, hyaluronic acid, and heparin can be used for soft tissue engineering. Goimil et al. [91] prepared and reported silk fibroin aerogel/poly(ε-caprolactone) scaffolds containing dexamethasone for bone regeneration. According to Li et al. [92], silk fibroin/gelatin nanoparticles aerogel is a promising bone tissue engineering material.

## 5. Blending

A new approach in biomedical science involves obtaining composite materials. Component materials usually have several disadvantages, including their mechanical properties or poor stability in an aqueous environment, because they swell and then dissolve [93,94]. It is, therefore, necessary to modify polymer-based materials by the addition of different natural or synthetic polymers [7,95]. In an aim to produce biomaterials characterized by unique structural and mechanical properties, with better physico-chemical properties, mixtures of two or more polymers can be used. The natural polymers, which are blend components, can be combined in the molten state (also known as melt mixing) [96] or they can be mixed as aqueous solutions in appropriate solvents [97,98,99]. If it is about melt mixing, the reaction between solids at high pressure and high temperature can be devastating to natural polymers, belonging to the protein group. The result of these high parameters may be denaturation and degradation of natural polymers [7]. The blending of biopolymers as aqueous solutions in appropriate solvents can be a solution of the above problem. However, some of the natural polymers are insoluble in common solvents. For this reason, miscibility studies can be desired and used. Four main groups of miscibility studies can be pointed out: methods based on the determination of optical homogeneity of the mixture; methods for the determination of glass transition temperatures; methods for the direct determination of interactions on molecular levels; indirect methods for the miscibility [7]. Fourier transform infrared spectroscopy (FTIR), viscometry and differential scanning calorimetry (DSC) are listed as the most common and easy techniques to investigate miscibility [7]. FTIR spectroscopy is used to study specific molecular bonding interactions in polymer blends. The changes in IR spectra (new bands, disappearance of some component bands, shifts in the specific bands) are characteristic for miscible systems. For immiscible blends, spectrum can reflect two individual components [7]. In viscometry method, experimental parameters of the mixture (*b*) and intrinsic viscosity (*η*) are comparing with their ideal (calculated) values, according to Krigbaum et al. [100] and Garcia et al. [101]. DSC is the most commonly used technique for determination of glass transition temperature (T_g_). Determination of the number of T_g_ is the main method for investigation of the number of amorphous phases in polymer systems. Each T_g_ corresponds to one amorphous phase and DSC provides determination of the number of the phases that coexist in a polymer mixture [7]. This part of the review is focused on silk fibroin complexes with other natural polymers used in biomaterial science. The peptide and polysaccharide additives are listed, and their influence on the properties of materials based on silk fibroin materials has been reported.

### 5.1. Collagen

Collagen (Coll) is one of the most abundant proteins in human and animal bodies. It is the main protein of the connective tissue and it is responsible for the strength of skin, tendons, cartilage tissue, and bones [7]. Silk fibroin/collagen materials as solutions, films and 3D scaffolds have been reported in the literature [37,95,97,102,103,104,105]. Miscibility of silk fibroin/collagen solutions has been studied [97]. The positive miscibility parameter for all the blends indicates good miscibility for all prepared blends (Table 2). This is due to the electrostatic interactions between chains and calcium ions from the solvent, which makes the whole mixture more stable [97].

Regarding silk fibroin/collagen films, it has been observed that mechanical properties were much better than for pure silk fibroin films [37]. Surface parameters were also different for silk fibroin/collagen mixture films than for pure silk fibroin films. Roughness increased with the addition of collagen into the silk fibroin matrix and the surface of mixed films was characterized by more hydrophobic characteristics (Table 3).

For skin tissue engineering, 50/50 and 75/25 silk fibroin/collagen porous membranes without and with TiO_2_ addition were studied [106]. It was found that 75/25 SF/Coll material without and with the addition of TiO_2_ was the best candidate for skin tissue regeneration. Silk fibroin/collagen 3D porous membranes were characterized by lower porosity and lower swelling degree than SF material. The reactions of *S. aureus* and *E. coli* bacteria and fibroblast cells to materials were studied. The materials inhibit the growth of bacteria and cell studies showed good biocompatibility and cell proliferation on fibroblasts [106]. Lin et al. prepared composite membranes of silk fibroin/collagen to use as material for cartilage repair in 50/50, 70/30, and 90/10 percentage fraction [104]. Cell proliferation was analyzed and the treatment of live/dead double staining simultaneous fluorescence staining of viable and dead cells was performed to assess the viability on chondrocytes (ATDC-5). Silk fibroin/collagen 70/30 film showed the suitable morphology, physical stability, and biological functionality to promote the proliferation of cells—a promising scaffold material for cartilage repair [104]. In turn, a double-layered collagen/silk fibroin composite scaffold was prepared by Wang et al. [107]. It was incorporated by TGF-β1/poly-L-lysine nanoparticles and the material was implanted into full-thickness articular cartilage defects of rabbits. It is expected that this composite can be a potential scaffold for cartilage tissue engineering material [107]. For silk fibroin/collagen 50/50 3D scaffolds, porosity, density, swelling ratio, and moisture content were investigated [108] (Table 4). It was found that SF/Coll materials had a lower density than pure biopolymer scaffolds, higher swelling ratio than silk fibroin material, and a higher amount of water than silk fibroin and collagen materials. The porosity was higher than silk fibroin and lower than collagen scaffold [108]. The microstructure of silk fibroin and silk fibroin/collagen 3D scaffolds was shown in Figure 8.

To promote bone regeneration, Apinun et al. [109] obtained silk fibroin and silk fibroin with 0.05% and 0.01% addition of collagen with a surfactant as a substance to gelation [109]. Collagen combined in the hydrogels yielded a positive effect on proliferation and matrix formation. However, authors expected more, and they suggested further studies to better understand and optimize the system as a scaffold and cell carrier in bone tissue regeneration [109]. On the other hand, materials-based silk fibroin/collagen 40/60 and 60/40 were good materials for bone tissue engineering in the research of Zeng et al. [110]. The in vitro cell proliferation using MG-63 cells was studied. The rate of degradation was steady, the pore sizes and porosity (above 90%) were suitable for the growth of osteoblasts [110].

### 5.2. Chitosan

Chitosan (CTS) is a polysaccharide, which is constructed of β-(1-4)-linked D-glucosamine (deacetylated unit) and N-acetyl-D-glucosamine (acetylated unit) (Figure 9). It can be obtained from chitin by the deacetylation process [7]. Chitosan is characterized by sufficient biological properties, and promotes normal tissue regeneration [111].

A lot of articles, where silk fibroin is mixed with chitosan can be found in the literature [99,110,112,113,114,115,116,117,118]. Miscibility was studied for 3 various weight ratios of silk fibroin/chitosan mixtures: 20/80; 50/50; 80/20 [99]. From the results it could be observed that silk fibroin and chitosan were miscible, except in the 20/80 SF/CTS mixture, where Δb_m_ was negative (Table 5). Miscibility is a result of a specific interaction between silk fibroin and chitosan. The same result was gained using the dynamic mechanical thermal analysis (DMTA). This method is very sensitive to the glass transition of a polymer system, which is the most general criterion for miscibility in polymer blends. Silk fibroin/chitosan blends were miscible when W_SF_ ≥ 50 [119].

Silk fibroin/chitosan-based films for the weight fraction of silk fibroin W_SF_ ≥ 50 were characterized by lower tensile strength (above 68 MPa for W_SF_ = 50 and above 66 MPa for W_SF_ = 80) than chitosan films (above 100 MPa). For materials with W_SF_ ≥ 20, Young’s modulus was higher (above 1.2 GPa for W_SF_ = 20; above 1.6 GPa for W_SF_ = 50; above 2.5 GPa for W_SF_ = 80) and tensile strain at break was lower (above 3.6% W_SF_ = 20; above 2.0% for W_SF_ = 50; above 0.4% for W_SF_ = 80) than pure chitosan films (Young’s modulus ≈ 0.9 GPa; tensile strain at brake ≈ 4.6%) [99]. Additionally, the dispersive component of surface free energy decrease, and polar component increase with an increasing amount of chitosan in the mixture. The surface is less rough for SF/CTS material than for SF scaffold [112]. Silk fibroin/chitosan 50/50 films can be used as a wound dressing and artificial skin. This is because of good mechanical properties and good water vapor and oxygen permeabilities [120]. Silk fibroin/chitosan 3D porous scaffolds were characterized by more than 90% of porosity, high water absorption, and high swelling ratio (up to 70.44 ± 1.13%) [110]. This is a good scaffold material for bone tissue engineering, especially SF/CTS 40/60 mixture because it has good biocompatibility and can promote the biological function of MG-63 cells [110]. According to other research, SF/CTS scaffolds were studied with 3T3 fibroblast cells, and sponges were biocompatible [118]. The best mechanical properties, taking into account the different compositions, were observed in SF/CTS 20/80 mixture [117]. The SF/CTS 50/50 scaffold was selected to test the inflammatory response in vivo. There was no obvious inflammatory response in vivo. After two and four weeks of observations, components and structure of the SF/CTS scaffold were beneficial for cell adherence, ingrowth, and the formation of new blood vessels [117]. This was the result for materials prepared by the lyophilization method. The electrospun SF/CTS nanofibrous membrane scaffolds for bone tissue engineering could be also fabricated by the electrospinning method [121]. The results of Lai et al. studies suggest that CTS and SF are excellent candidates for proliferation and osteogenic differentiation of hMSCs (human bone marrow mesenchymal stem cells), respectively. A SF/CTS blend preserved the osteogenesis nature of CTS without diminishing the cell proliferative effect of SF [121]. Sun et al. [122] decided to compare SF/Coll and SF/CTS scaffolds. SF/Coll showed better characteristics, and they claim that these mixtures can be served as new materials for scaffolds for cartilage and bone tissue engineering [122].

### 5.3. Alginate

Alginate (AL) is a polysaccharide of linear chain composed of β-D-mannuronic acid and α-L-glucuronic acids (Figure 10). It can be extracted from brown algae. Alginate and its salts, e.g., sodium alginate (SA) are promising natural substances to use in blends with silk fibroin [123]. In recent years some articles about silk fibroin/sodium alginate materials have been published [123,124,125,126,127,128,129].

Miscibility and physical characteristic were described by de Moraes et al. [123]. It was found that blends where an aqueous solution of sodium alginate was used were heterogeneous and the blending solutions did not mix well. The same result was observed for all belonging ratios. For this reason the authors decided to use an alkaline solution of sodium alginate. It was better because no macroscopic phase separation was detected. For further research, 25/75 SF/SA mixture was used, because only for this weight ratio, a film without silk fibroin fibril formation could only be obtained. Silk fibroin/sodium alginate 25/75 film was characterized by better mechanical properties than pure silk fibroin films. Additionally, higher swelling capacity, good permeability allowing fluid exchange with the environment, and non-cytotoxicity were observed for two-component films [123]. Biodegradable silk fibroin microsphere modified with sodium alginate were fabricated and studied to use as potential biocompatible and biodegradable embolic agents for transcatheter arterial embolization (TAE) [130]. The microspheres had a porous structure, spherical shape, and desirable particle size for the injection through a catheter for TAE treatment. Additionally, hemolysis assay in vitro showed the microspheres exhibited good blood compatibility, cytotoxicity analysis showed almost had no cytotoxicity on the cells [130]. Silk fibroin/sodium alginate composite porous scaffolds were observed by in vitro enzymatic degradation in Collagenase IA [131]. The degradation rate of the composites can be controlled by changing the blend ratio. SF/SA 50/50 kept integrating pore structure after degrading for 18 days. As the degradation time increased, the weight of the composites decreased. It was also observed that more than half of the SF/SA 3D porous scaffold composite were degraded after subcutaneous implantation in Sprague Dawley rats for three weeks, meanwhile, these materials were well tolerated by the animals [131]. Silk fibroin/sodium alginate films for use as promising wound dressing material with excellent cytocompatibility and proangiogenesis action for wound healing were also researched [132]. Biomimetic SF/SA composite scaffolds for soft tissue engineering were characterized by regular and uniform pore morphology, promoting cellular attachment and proliferation for tissue regeneration [133]. Regarding SF/SA hydrogels, they degraded quickly after incubating in protease XIV solution than in PBS solution at 37 °C, compressive stress decreased slightly with increasing of sodium alginate content, and after 12 day of cultivating, human mesenchymal stem cells proliferated [134].

### 5.4. Hyaluronic Acid

Hyaluronic acid (HA) is a glycosaminoglycan which belongs to the polysaccharide group. It is present in the body as the main component of the cellular matrix. It reveals hydration capacity and excellent hygroscopic properties [111]. Hyaluronic acid is composed of D-glucuronic acid and N-acetyl-D-glucosamine, linked via alternating β-(1-4) and β-(1-3) glycosidic bonds (Figure 11).

Combining the properties of natural protein and polysaccharides can be a promising strategy to obtain bioactive materials with a controlled structure in the biomedical field [135,136,137,138,139,140,141,142,143,144,145]. In silk fibroin/hyaluronic acid scaffolds, increasing hyaluronic acid content significantly enhanced the water binding capacity and flexibility [140]. The flexibility and water-absorbing quality of silk fibroin/hyaluronic acid hydrogels were improved due to the hydrophilicity of hyaluronic acid addition [140]. However, high amounts of hyaluronic acid in mixtures reduced the water stability [141]. The silk fibroin/hyaluronic acid hybrid hydrogel exhibited gel-sol transition under shear stress, which suggests potential application as an injectable material [142]. The studies on fibroblast vitality and morphology showed that silk fibroin/hyaluronic acid hydrogels can support cell adhesion, differentiation, proliferation in vitro, and present better biocompatibility in comparison to pure SF hydrogels [140]. In in vivo studies, the hydrogels based on silk fibroin/hyaluronic acid blends presented good histocompatibility and promoted vascular-like tissue regeneration when were implanted subcutaneously in Sprague Dawley rats [143]. Regarding microspheres to the application on controlled release, the microspheres having higher silk fibroin content showed slower degradation rate and more stability [144]. Curcumin, used as an antioxidant with anticancer activity, was adsorbed in the microspheres and all microspheres released curcumin in a controllable manner. This was possible because of the hydrophobic interaction between curcumin and the crystalline domain of silk fibroin [144]. The silk fibroin/hyaluronic acid porous scaffolds have promise as a dermal substitute because results of Zhang et al. in vitro studies showed that scaffolds could support the fibroblast cell adhesion and proliferation and showed good cytocompatibility [145]. The pore radius and porosity decreased with a decrease in the freezing temperature and increase in the hyaluronic acid ratio. In vivo, after implantation of silk fibroin/hyaluronic acid scaffolds to Sprague Dawley rats, a new dermal layer was formed, as determined by histological analysis [145].

### 5.5. Other Polymers

Silk fibroin can be mixed also with other polymers (Table 6). It can be mixed with natural polymers, as in the examples described above, and with synthetic polymers. In addition to biopolymers described above, the group of natural polymers with which the mixtures of silk fibroin were tested are gelatin [146,147,148], cellulose [149,150], agarose [151], keratin [152], elastin [153], chitin [154], heparin [155], and carrageenan [156]. They can be obtained as liquid states, films, 3D porous sponges, particles, and hydrogels, and they can be used in the broadly understood biomedical field. The same is true for synthetic polymers. These can be mixed with silk fibroin and can create various structures: liquid states, membranes, scaffolds, hydrogels, micro, and nanoparticles. Silk fibroin-based mixtures with poly(ethylene glycol) [157], poly(vinyl alcohol) [158], polyacrylamide [159], polycaprolactone [160], poly(lactic-co-glycolic acid) [161], polyurethane [162] and polylactide [163] have been reported.

### 5.6. Three-Component Blends

To improve the specific parameters useful in tissue engineering, three components can be blended (Table 7). Many three-component mixtures with the presence of natural polymers have already been investigated: carrageenan/chitosan/gelatin [164], alginate/chitosan/collagen [165], or chitosan/collagen/hyaluronic acid [98,166,167]. This review is describing silk fibroin-based materials. There are many papers regarding triple blends with the presence of silk fibroin. Table 7 summarizes the compositions of ternary mixtures with the presence of silk fibroin, which can be found in the literature.

### 5.7. Inorganic Additives

Biocomposites are materials containing biopolymers and ceramics, for example, titanium dioxide, hydroxyapatite, or magnetic particles [111]. They can be named the ‘hard’ phase [7]. Silk fibroin materials with the addition of inorganic additives can be characterized by biocompatibility, biodegradability, and good mechanical properties. The degradation process of materials containing inorganic additives is longer, and materials are not easily resorbed in a short time. There is not easy to obtain polymeric material with inorganic particles, because it should be characterized by three main features, they must: be non-toxic; have excellent mechanical properties to fulfill their role as support for tissue reconstruction; and have the degradation rate proper for the tissue regeneration ability [32,111]. One, two, or three-component materials with the addition of inorganic particles have been reported (Table 8). Several research groups are working on the development of new polymeric biomaterials on the blends, which contain inorganic particles [7]. This kind of new material can be used as a biomaterial in hard and soft tissue applications. Materials with additives are prepared by the incorporation in the polymeric matrix. Hydroxyapatite (Ca_10_(PO_4_)_6_(OH)_2_; HAp) is the major inorganic component of natural bone [32]. It is biocompatible, bioactive, osteoconductive, non-toxic, and non-immunogenic. It is widely used in orthopedic and dental materials because of its excellent bioactive and biocompatibility abilities [32,111]. Hydroxyapatite was successfully added or precipitated in silk fibroin-based matrices [95,177,178,179,180]. Titanium dioxide (TiO_2_) is characterized by light weight and resistance toward corrosion and it is frequently used in the fabrication of bone implants [181]. The addition of titanium dioxide to the polymeric matrix favorably influences adhesion, proliferation, and differentiation of osteoblast [181]. Physico-chemical characterization of polymeric materials with the addition of titanium dioxide has been reported [106,181,182,183,184,185]. Bioactive glass, zinc oxide, and magnetic particles can also be added also to silk fibroin-based materials [7,111]. Biocompatibility, bioactivity, and osteoconductive characterize bioactive glasses. Silk fibroin, silk fibroin/chitosan, silk fibroin/poly(vinyl alcohol), alginate-poloxamer/silk fibroin composites with the presence of bioactive glasses have been reported as nanoparticles, films, hydrogels, and scaffolds to use in biomaterials applications [186,187,188,189,190,191]. Zinc oxide particles have received growing interest as a promising material, because of their non-toxicity, cost-effectiveness, and high adsorption capacity [192]. Chitosan/silk fibroin/zinc oxide nanocomposites, silk fibroin-modified disulfiram/zinc oxide nanocomposites, and hyaluronic acid-based silk fibroin/zinc oxide dressing have been described in the literature [192,193,194]. Magnetite particles are characterized by high affinity to many chemical substances and chemical stability. They are biocompatible, biodegradable, non-toxic, and thanks to the above properties, they are commonly used in biomedical applications (tissue repair, drug delivery, cellular therapy) [195]. There are very few reports of silk fibroin materials with the presence of magnetic particles: magnetic silk fibroin composite nanofibers, magnetic silk fibroin e-gel scaffolds, silk fibroin films decorated with magnetic particles, and silk fibroin/chitosan/magnetite scaffolds have been studied [196,197,198,199]. Silk fibroin-based materials with addition of noble metals (tetrachloropalladate (II) (Na_2_PdCl_4_), potassium tetrachloroplatinate (II) (K_2_PtCl_4_), gold chloride trihydrate (HAuCl_4_·3H_2_O), silver nitrate (AgNO_3_), tetrachloroauric acid (HAuCl_4_)) also can be found in the literature [200,201]. They can be used for biological sensing, energy storage applications, and in biotechnology such as immunoassay, biosensor, DNA identification and inspection, gene therapy, and will emerge in further research [200,201].

## 6. Cross-Linking

Modification of silk fibroin, by addition of second and third components, was described above. In addition, such mixtures can be subjected to the cross-linking process, to improve several parameters, for example, stability in water conditions and regularity of pores. One component material and non-cross-linked blended material can be characterized by non-elastic and poor stability in an aqueous environment (they swell and then dissolve) [93,94]. After the cross-linking process, biomaterials should not be affected by changes in biodegradability and biocompatibility. From this type of modification, it is expected to improve the mechanical properties of materials, stability in water conditions and to improve degradation resistance [205]. Due to the availability of amine and acid side chains, silk fibroin can be easily modified to suit a wide range of biomedical field applications [34]. Crosslinking is the most common approach to modifying polymeric materials. Three types of cross-linking can be classified: enzymatic, physical, and chemical cross-linking [206,207,208] (Figure 12).

The mechanisms of various cross-linking processes are shown in Figure 13. The advantages and disadvantages of three kinds of cross-linking are highlighted in Table 9.

The modification and crosslinking of polymers and biopolymers using enzymes has captured high interest among research groups [207]. Reactions with enzymes are characterized by high specificity, lack of side products, and low energy demand. Transglutaminase, tyrosinase, and horseradish peroxidase can be used for enzymatic cross-linking [208,210]. A physical cross-linking modification usually leads to the change of physico-chemical properties of the polymer through the action of physical factors. This kind of cross-linking includes different techniques, e.g., gamma radiation, laser treatment, UV irradiation, dehydrothermal treatment [206,207,208]. Physical modification is a simpler and cheaper method compared to other processes of polymer modification [211]. Chemical cross-linking is the process used to covalently bridge polymeric chains to improve polymer properties [207,212] The chemical-cross-linking method is considered to be the most effective and predictable [205]. It provides for formation of very strong bonds. The disadvantages of chemical cross-linking is the high price (chemical cross-linking is more expensive than physical cross-linking), necessity to toxicity testing and necessity to remove the residual of cross-linker [209]. Some detoxifying strategies have been proposed to exclude toxic effects of cross-linker residual. For instance, if the cross-linker has aldehyde groups (e.g., glutaraldehyde), washing of cross-linked scaffolds with solutions containing free amino groups or amino acid solutions (e.g., glycine) can be used which leads to the removal of free aldehyde groups [207,209]. Regarding carbodiimide agents (e.g., N-(3-dimethylaminopropyl)-N’-ethylcarbodiimide hydrochloride—EDC), all of the residues are water soluble and they can be washed out of the cross-linked scaffold construct easily by distilled water after the completion of the cross-linking reaction [207]. The concentration of cross-linking agent also has a big influence on the toxicity effect of the cross-linking process [213]. For example, concentration up to 8% of glutaraldehyde has shown no toxicity for cross-linking [209]. According to literature, the fabrication process is also very important [214]. For instance, glutaraldehyde can be added to collagen/chitosan material before the freeze-drying procedure to obtain better properties after cross-linking process, however the toxicity effect of glutaraldehyde in this manner can be higher than during the protocol when glutaraldehyde was added after the freeze-drying stage [209,214]. To sum up, the type and amount of material, concentration of cross-linker and fabrication protocol affect the biocompatibility of material [207,209]. There are a lot of cross-linking agents used for chemical cross-linking [206,207,208], e.g., glutaraldehyde [215], genipin [59,179], dialdehyde starch [108], glyoxal [171], EDC (N-(3-dimethylaminopropyl)-N’-ethylcarbodiimide hydrochloride, NHS (N-hydroxysuccinimide), EDC/NHS mixture [102,172,180]. The structural formulas of some chemical cross-linkers are placed in Figure 14.

## 7. Conclusions

This review can be used by wide group of scientists and researchers, who are working to obtain appropriate material useful in broadly understood biomedicine. The methods of physico-chemical and biological properties silk-based materials improvement were reported and discussed. Blending with other polymers and cross-linking can be used to improve silk fibroin-based materials properties, which can be desirable in the biomedical field, especially in tissue engineering. Blending includes natural and synthetic polymers, cross-linking includes enzymatic, physical, and chemical processes. There are many articles regarding silk fibroin and complexes based on silk fibroin modified by the addition of second and third polymer or cross-linking process and it is hard to compare results reported by different research groups. The published results are rather consistent in the presented papers. Variability may result from the fact that working with polymers obtained from natural sources is not easy. The physico-chemical properties of obtained silk fibroin are a little bit different in each batch of cocoons or source material. In addition, as for the results of mixtures of silk fibroin with polysaccharides, the differences in properties are probably caused by differences in the polymer molecular weight and deacetylation degree.

Some features make silk fibroin a promising base to obtain biomaterials for many clinical functions: the unique structure, biocompatibility, versatility in processing, availability of different morphologies (fibers, films, 3D porous structures, particles, hydrogels), options for genetic engineering of variations of silks, thermal stability, the ease of sterilization, surface chemistry for facile chemical modifications, and controllable degradation. Each of the silk fibroin-based systems has shown promising features for different biomedical applications. More research will have to be done before silk fibroin can be used for clinical trials and commercialized for tissue engineering applications, especially for bone tissue engineering because there are few in vivo studies with silk fibroin materials and these studies were done mostly with small animals (e.g., rats) that do not sufficiently predict their performance in humans. A better understanding is needed regarding silk fibroin systems to create tissues which will be able to remodel similar to bone. The long-term biocompatibility, biodegradability, and degraded products, along with the ability to tune silk morphologies for tissue-specific requirements also need to be better understood. It can be expected that with novel processing techniques, new silk fibroin-based composites will be developed in the future and open up even more possibilities for tissue engineering applications. The future for silk fibroin biomaterials to impact clinical needs appears promising and has potential to bring about viable strategies and innovations.

## Figures and Tables

**Figure 1 materials-14-01510-f001:**
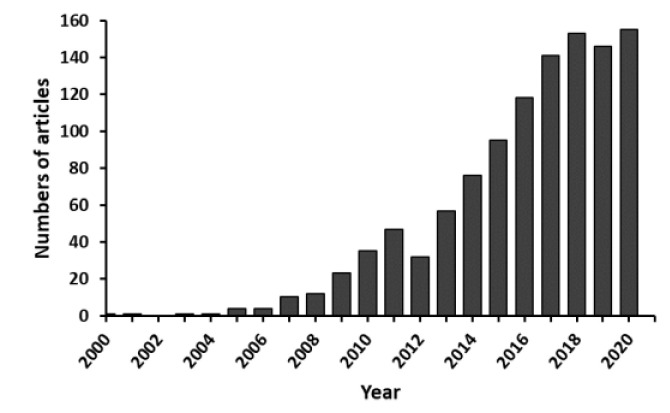
Annual publications on silk fibroin in tissue regeneration since 2000–2020. The search engine was Web of Science. “Silk fibroin” and “tissue regeneration” were applied as the terms for searching.

**Figure 2 materials-14-01510-f002:**
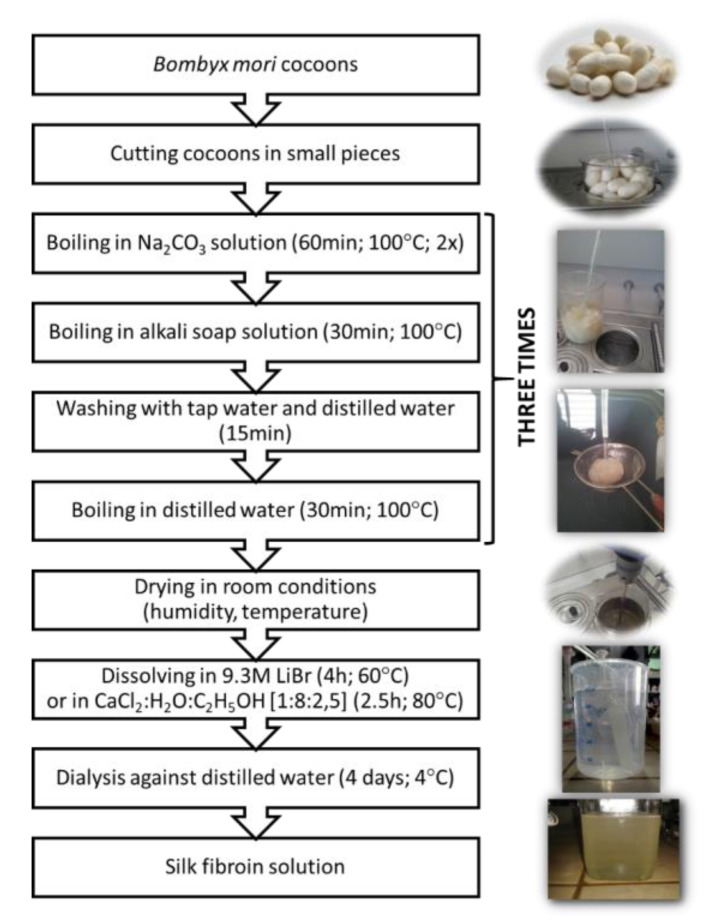
Scheme of *Bombyx mori* silk fibroin obtaining.

**Figure 3 materials-14-01510-f003:**
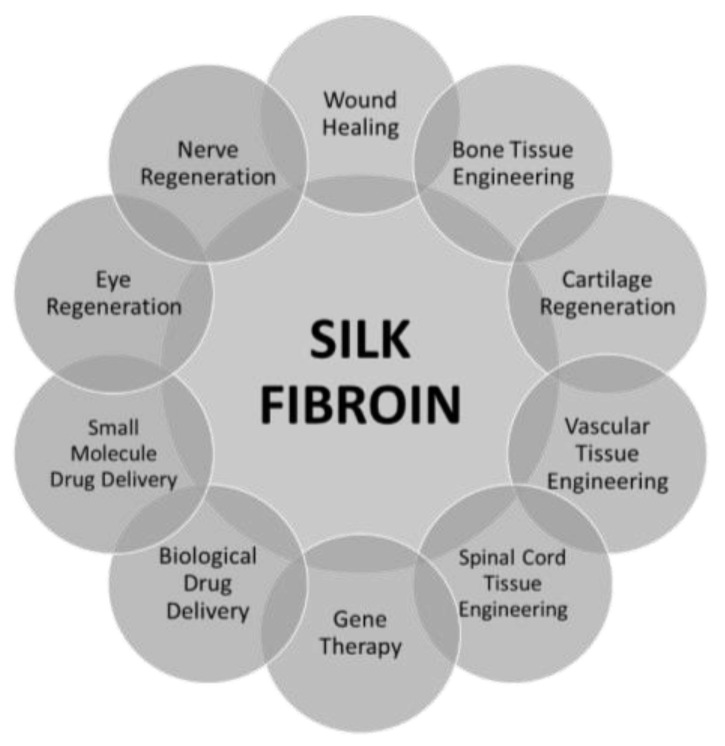
Schematic of various silk fibroin applications.

**Figure 4 materials-14-01510-f004:**
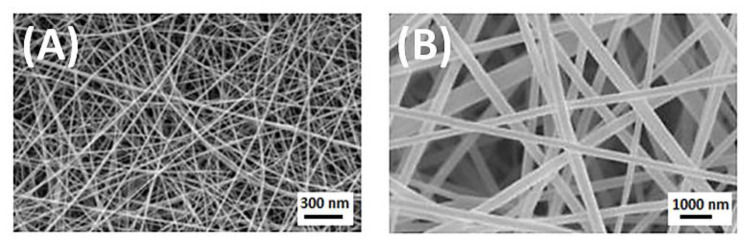
(**A**) Silk fibroin fiber obtained by spinning method with an average diameter of 65.1 nm; (**B**) Scanning Electron Microscope (SEM) image of electrospun silk fibroin nanofibers with diameter 400 ± 76 nm. Reprinted from [54] with permission from Elsevier.

**Figure 5 materials-14-01510-f005:**
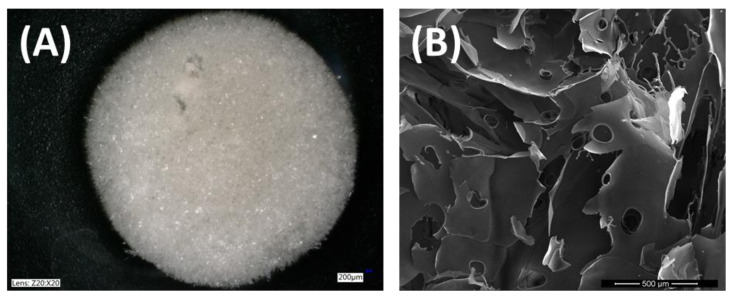
Silk fibroin scaffolds obtained by lyophilization method (freeze dryer, ALPHA 1–2 LDplus, CHRIST, −20 °C, 100 Pa, 48 h), (**A**) photograph from optical microscope (Keyence VHX-900F); (**B**) SEM image with magnification 500 µm (LEO Electron Microscopy Ltd., England, UK).

**Figure 6 materials-14-01510-f006:**
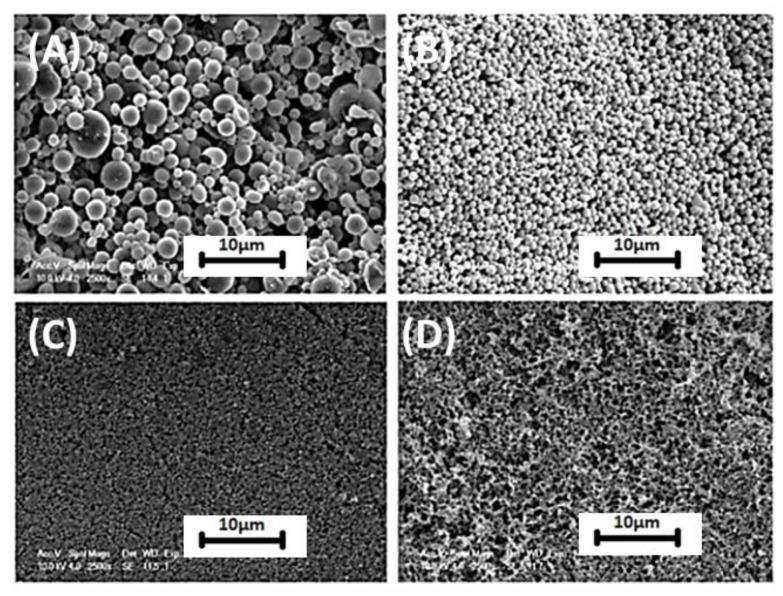
Silk particles fabricated with addition of PVA (V_PVA_ = 2V_silk + ethanol_): (**A**) V_silk_/V_ethanol_ = 20; (**B**) V_silk_/V_ethanol_ = 5; (**C**) V_silk_/V_ethanol_ = 2.5; (**D**) silk sedimentation formed with addition of PVA, V_silk_/V_ethanol_ = 0.5. Reprinted from [68] with permission from Elsevier.

**Figure 7 materials-14-01510-f007:**
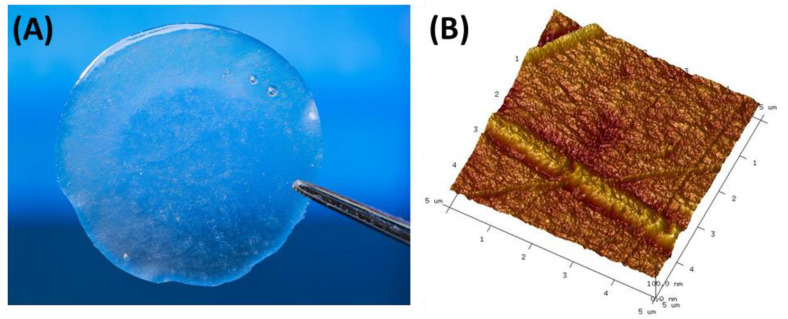
**(A)** Silk fibroin-based film obtained by casting method and (B) Atomic Force Microscopy (AFM) picture of silk fibroin-based film acquired at fixed resolution (512 × 512 data points) using scan width 5 μm with a scan rate of 1.97 Hz (microscope with a Nanoscope IIIa controller, Digital Instruments, Santa Barbara, CA, USA).

**Figure 8 materials-14-01510-f008:**
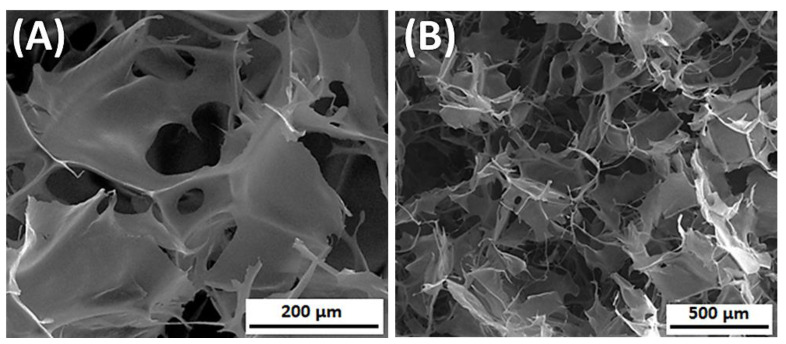
The SEM images of silk fibroin/collagen 3D porous scaffolds, prepared by lyophilization (SF dissolved in 9.3M LiBr), (**A**) magnification 500×; (**B**) magnification 150×.

**Figure 9 materials-14-01510-f009:**
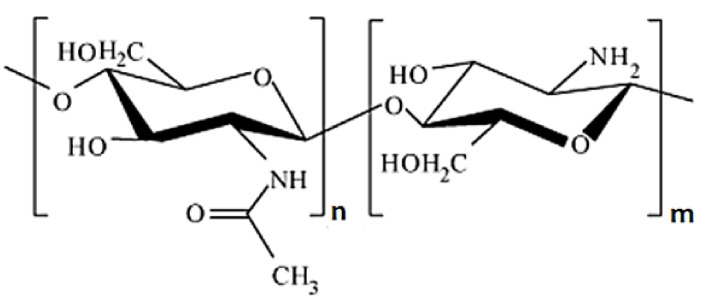
The chitosan structure.

**Figure 10 materials-14-01510-f010:**
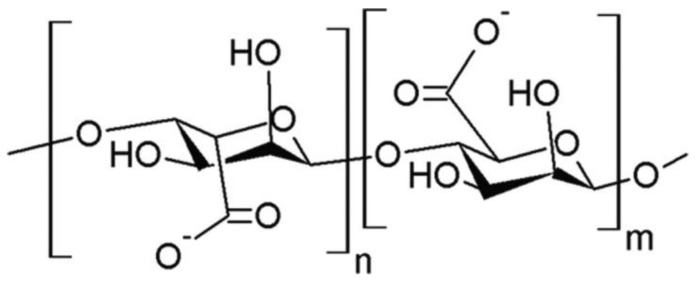
The alginate structure.

**Figure 11 materials-14-01510-f011:**
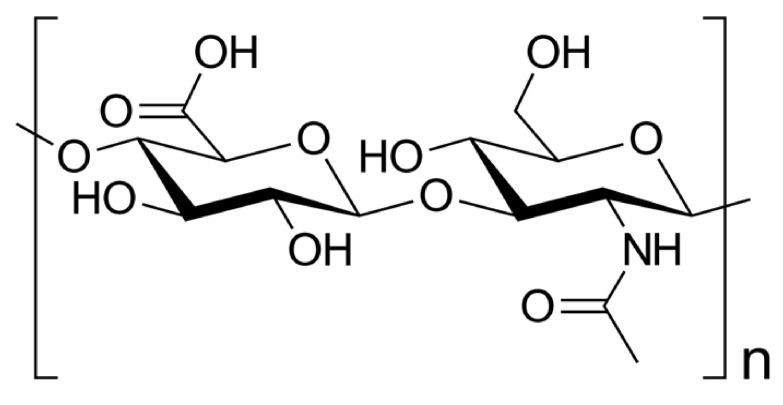
The hyaluronic acid structure.

**Figure 12 materials-14-01510-f012:**
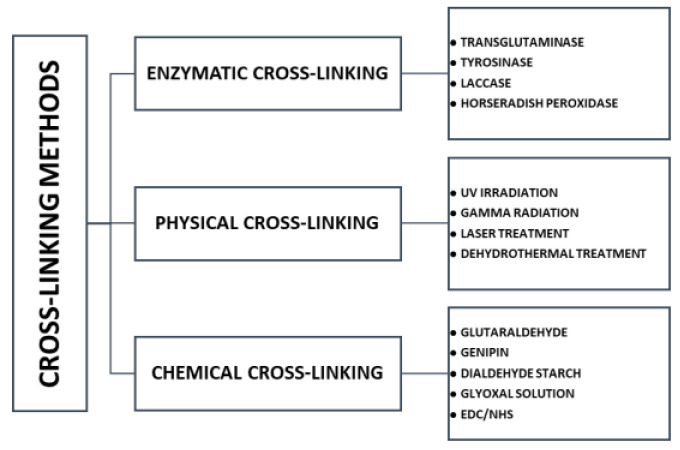
The scheme of cross-linking classification with typical examples.

**Figure 13 materials-14-01510-f013:**
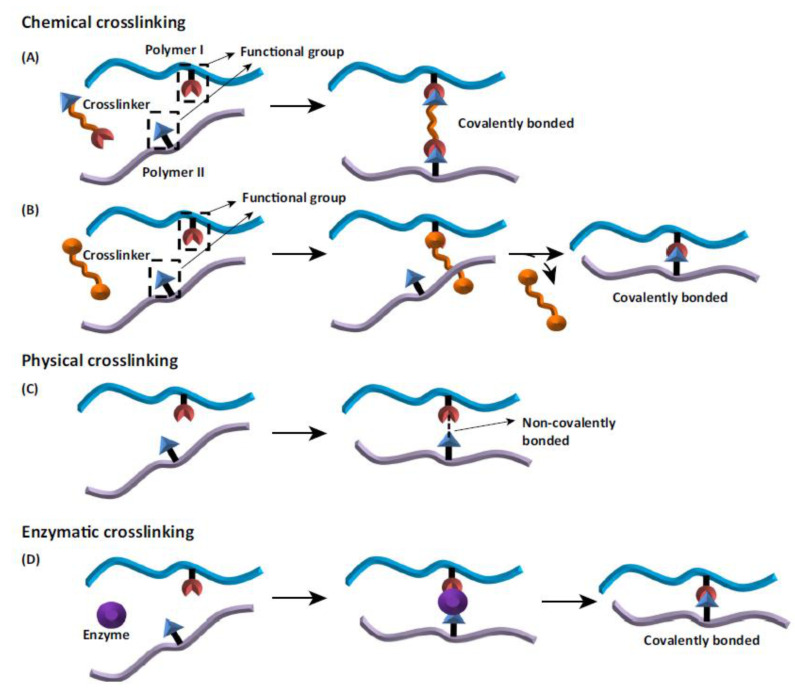
Schematic of the various mechanisms of crosslinking: (**A**) chemical cross-linker with the polymer chains incorporated into the bond; (**B**) chemical cross-linker leaves the reaction after cross-linking process; (**C**) physical cross-linkers form a non-covalent bond between the polymer chains; (**D**) enzymatic crosslinking method. Reprinted from [206] with permission from Elsevier.

**Figure 14 materials-14-01510-f014:**
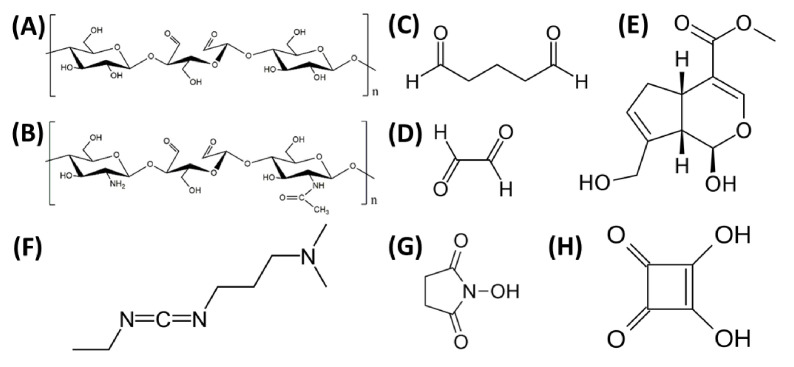
The structural formulas of commonly cross-linkers: (**A**) dialdehyde starch; (**B**) dialdehyde chitosan; (**C**) glutaraldehyde; (**D**) glyoxal; (**E**) genipin; (**F**) 1-Ethyl-3-(3-dimethylaminopropyl) carbodiimide hydrochloride (EDC); (**G**) N-hydroxysuccinimide (NHS); (**H**) squaric acid.

**Table 1 materials-14-01510-t001:** The most relevant types of morphologies with processing methods and applications.

Morphology Type	Processing Method	Applications	Source
fibers	reeling, spinning, electrospinning	tissue engineering, wound healing, nerve guides, stents and drug delivery systems	[34,52,53,54]
3D structures (scaffolds, sponges, foams)	lyophilization, particulate-leaching, gas foaming, electrospinning, 3D printing	bone tissue engineering, cartilage tissue engineering, drug delivery, skin tissue engineering, intervertebral disc, meniscus liver	[24,54,55]
particles (microspheres, nanolayers, nanocoatings, nanoparticles, microcapsules)	spry drying, water in oil (W/O) emulsion solvent evaporation, water in oil emulsion solvent diffusion, ball-milling method, milling, freeze-drying and grinding, spray drying, self-assembly, freeze-thawing, jet breaking, desolvation, PVA blending	controlled-release drug delivery systems, bioactive molecule delivery, vaccine delivery, tissue engineering	[57,58,59,60,61,62,63,64,65]
films	casting, Langmuir-Blodgett (LB) process, spin coating, vertical deposition, manual or spin assisted layer by layer assembly, centrifugal casting	wound dressing/skin repair, biosensors, coating materials, bone, corneal, vascular tissue engineering	[53,64,74,75,76,77,78,79]
hydrogels	sol-gel transition in the presence of acid, ions or other additives, sonication, lyophilization	cartilage tissue engineering, guided bone repair, drug release/delivery	[41,53,65,80,81,82]
aerogels	ambient drying, freeze drying, direct supercritical drying, supercritical drying by CO_2_ extraction.	nerve regeneration, soft tissue engineering, bone tissue engineering,	[87,89,90,91,92]

**Table 2 materials-14-01510-t002:** Theoretical (by Krigbaum and Wall [100] and Garcia et al. [101] methods) and experimental values silk fibroin/collagen mixtures. Adapted from Ghaeli I. et al., Phase Behavior and Miscibility Studies of Collagen/Silk Fibroin Macromolecular System in Dilute Solutions and Solid State [97].

W_SF_/W_Coll_	b^exp^_m_ [dL/g]^2^	b^id*^_m_ [dL/g]^2^	∆*b_m_*_*_	b^id**^_m_ [dL/g]^2^	∆*b_m_*_**_	Miscibility
25/75	72.41	30.04	42.37	21.64	50.77	✔
50/50	75.91	20.87	55.04	9.67	66.24	✔
75/25	39.54	10.94	28.60	2.54	37.00	✔

W_SF_—percentage fraction of silk fibroin in mixture; W_Coll_—percentage fraction of collagen in mixture; b—polymer–polymer interactions term at finite concentration related to the Huggins coefficient; b^id^*_m_: determined according to Krigbaum and Wall; b^id^**_m_: determined according to Garcia et al.; *Δb_m_* = b^exp^_m_ − b^id^_m_.

**Table 3 materials-14-01510-t003:** Parameters describing the mechanical properties and surface of silk fibroin/collagen films. Adapted from Sionkowska A. et al., Polymer films based on silk fibroin and collagen—the physico-chemical properties [37].

%_SF_	Contact Angle [^o^]		γ_S_ [mJ/m^2^]	γ_S_^P^/ γ_S_^D^	E_mod_ [GPa]	F_max_ [MPa]	R_q_ [nm]
	D	GL					
100	51.4 ± 0.3	88.7 ± 0.4	33.13	0.026	0.43 ± 0.06	4.12 ± 0.94	5.10
90	50.8 ± 0.1	89.4 ± 0.2	33.55	0.020	0.92 ± 0.26	4.99 ± 1.34	5.90
70	49.6 ± 0.1	85.0 ± 0.3	33.97	0.046	1.02 ± 0.30	9.63 ± 2.51	7.60
50	49.9 ± 1.8	82.4 ± 0.7	33.82	0.071	0.82 ± 0.52	21.9 ± 10.7	6.70
30	48.0 ± 0.2	83.6 ± 0.3	34.83	0.050	0.79 ± 0.15	1.70 ± 1.06	7.40
10	44.3 ± 0.8	80.9 ± 0.3	36.80	0.058	2.55 ± 0.41	66.1 ± 4.93	33.00
0	44.4 ± 0.2	80.4 ± 0.4	36.76	0.063	2.41 ± 0.74	81.2 ± 1.87	26.60

%_SF_—percentage fraction of silk fibroin in mixture; D—diiodomethane; GL—glycerol; γ_S_—surface free energy; γ_S_^P^/ γ_S_^D^—the ratio of polar to dispersive components of surface free energy; E_mod_—Young’s modulus, F_max_—tensile strength; R_q_—roughness parameter.

**Table 4 materials-14-01510-t004:** The result of porosity, density, swelling ratio and moisture content of silk fibroin, collagen and their 50/50 mixture based scaffolds. Adapted from Grabska-Zielińska et al., Physico-Chemical Characterization and Biological Tests of Collagen/Silk Fibroin/Chitosan Scaffolds Cross-Linked by Dialdehyde Starch [108].

Sample	Porosity [%]	Density [mg/cm^3^]	Swelling Ratio [%]	Moisture Content in 100 g of Dry Sample [g]
SF	95 ± 1.7	33.0 ± 0.4	1511 ± 147	7.09 ± 1.03
Coll	88 ± 0.5	16.9 ± 2.2	^Nd^	14.17 ± 1.36
SF/Coll	97 ± 1.0	14.8 ± 2.1	1891 ± 236	19.27 ± 0.92

^Nd^—not determined.

**Table 5 materials-14-01510-t005:** Theoretical (by Krigbaum and Wall [100] and Garcia et al. [101] methods) and experimental values silk fibroin/chitosan mixtures. Adapted from Sionkowska A. et al., Miscibility and physical properties of chitosan and silk fibroin mixtures [99].

W_SF_/W_CTS_	b^exp^_m_ [dL/g]^2^	b^id*^_m_ [dL/g]^2^	∆*b_m_*_*_	b^id**^_m_ [dL/g]^2^	∆*b_m_*_**_	Miscibility
20/80	115	147	−32	147	−32	×
50/50	58	58	0	57	1	✔
80/20	36	10	26	9	27	✔

W_SF_—percentage fraction of silk fibroin in mixture; W_CTS_—percentage fraction of chitosan in mixture; b—polymer–polymer interactions term at finite concentration related to the Huggins coefficient; b^id*^_m_: determined according to Krigbaum and Wall; b^id**^_m_: determined according to Garcia et al.; *Δb_m_* = b^exp^_m_ − b^id^_m_.

**Table 6 materials-14-01510-t006:** List of silk fibroin-based composite materials with natural and synthetic polymers to use in the biomedical field.

Scaffold Composition	Reference
**Natural polymers**	
Silk Fibroin + Gelatin	[146,147,148]
Silk Fibroin + Cellulose	[149,150]
Silk Fibroin + Agarose	[151]
Silk Fibroin + Keratin	[152]
Silk Fibroin + Elastin	[153]
Silk Fibroin + Chitin	[154]
Silk Fibroin + Heparin	[155]
Silk Fibroin + Carrageenan	[156]
**Synthetic polymers**	
Silk Fibroin + Poly(ethylene Glycol) (PEG)	[157]
Silk Fibroin + Poly(vinyl Alcohol) (PVA)	[158]
Silk Fibroin + Polyacrylamide (PAM)	[159]
Silk Fibroin + Polycaprolactone (PCL)	[91,160]
Silk Fibroin + Poly(Lactic-co-glycolic Acid) (PLGA)	[161]
Silk Fibroin + Polyurethane (PUR)	[162]
Silk Fibroin + Polylactide (PLA)	[163]

**Table 7 materials-14-01510-t007:** The different silk fibroin-based materials compositions with their morphology.

Composition	Morphology	Reference
Silk Fibroin + Chondroitin Sulphate + Hyaluronic Acid	3D scaffold	[21]
Silk Fibroin + Polylactide + Gelatin	3D scaffold	[50]
Silk Fibroin + Poly(lactic-co-glycolic Acid) + Collagen	3D scaffold	[50]
Silk Fibroin + Nanochitosan + Hyaluronic Acid	3D scaffold	[168]
Silk Fibroin + Polyethylene Glycol + Keratin	3D scaffold	[169]
Silk Fibroin + Hyaluronic Acid + Sodium Alginate	3D scaffold	[21,170]
Silk Fibroin + Hyaluronic Acid + Heparin	3D scaffold	[90]
Silk Fibroin + Chitosan + Collagen	3D scaffold	[108,171,172]
Silk Fibroin + Chitosan + Polylactide	3D scaffold	[50]
Silk Fibroin + Chitosan + Heparin	3D scaffold	[50]
Silk Fibroin + Collagen + Hyaluronan	3D scaffold	[50]
Silk Fibroin + Collagen + Heparin	3D scaffold	[173]
Silk Fibroin + Β-Cyclodextrin + Polyethyleneimine	hydrogel	[21]
Silk Fibroin + Calcium Alginate + Carboxymethyl Cellulose	hydrogel	[21]
Silk Fibroin + Chitosan + Alginate Dialdehyde	film	[174]
Silk Fibroin + Chitosan + Polylactide	film	[175]
Silk Fibroin + Cellulose + Chitin	fibers	[176]
Silk Fibroin + Poly(caprolactone) + Hyaluronic Acid	nanofibrous matrices	[21]
Silk Fibroin + Hyaluronic acid + Heparin	aerogel	[90]

**Table 8 materials-14-01510-t008:** The list of polymeric-based materials with presence of inorganic additives.

Inorganic Additives	Material	Reference
Hydroxyapatite	Silk Fibroin/Hydroxyapatite Silk Fibroin/Nanohydroxyapatite Silk Fibroin/Collagen/Hydroxyapatite Silk Fibroin/Collagen/Nanohydroxyapatite Silk Fibroin/Chitosan/Nanohydroxyapatite Hydroxyapatite/Gelatin/Silk Fibroin Poly (vinyl Alcohol) /Silk Fibroin/Nano-Hydroxyapatite Silk Fibroin/Titanium Dioxide/Hydroxyapatite Hydroxyapatite/Silk Fibroin/Polycaprolactone Silk Fibroin/Nano-Hydroxyapatite/Polyethylene Glycol Silk Fibroin-Alginate-Hydroxyapatite	[50,95,177,178,179,180,202,203,204]
Titanium dioxide	Silk Fibroin/Titanium Dioxide Silk Fibroin/Titanium Dioxide/Hydroxyapatite Chitin/Silk Fibroin/Titanium Dioxide Silk Fibroin/Collagen/Titanium Dioxide	[106,181,182,183,184,185]
Bioactive glass	Silk Fibroin/Bioactive Glass Silk Fibroin/Poly (vinyl Alcohol) /Bioactive Glass Chitosan/Silk Fibroin/Bioactive Glass Bioactive Glass/Chitosan/Silk Fibroin Bioactive Glass/Alginate-Poloxamer/Silk Fibroin	[186,187,188,189,190,191]
Zinc oxide	Chitosan/Silk Fibroin/Zinc Oxide Hyaluronic Acid/Silk Fibroin/Zinc Oxide Silk Fibroin/Disulfiram/Zinc Oxide	[192,193,194]
Magnetic particles	Magnetic Silk Fibroin Composite Magnetic Silk Fibroin E-Gel Scaffolds Silk Fibroin Films Decorated With Magnetic Particles Silk Fibroin/Chitosan/Magnetite Scaffolds	[196,197,198,199]
Noble metals	Silk–Palladium Aerogel Fibers Silk-Platinum Aerogel Fibers Silk fibroin-Gold Colloid Silk fibroin-Silver Colloid	[200,201]

**Table 9 materials-14-01510-t009:** The advantages (+) and disadvantages (**−**) of enzymatic, physical and chemical cross-linking. Adapted from [207,208,209].

**(+)**	**(−** **)**
**Enzymatic cross-linking**	
Unlike many chemical agents, enzymes are most active under mild aqueous reaction conditions Crosslinking process can often be controlled by modifying temperatures, pH, or ionic strength	The most expensive crosslinker Substrate specificity
**Physical cross-linking**	
Safe Less toxic for cells than chemical agents Inexpensive Minimum tissue reaction after crosslinking process	Bonds are weaker than the chemical crosslinkers May alter the properties of the materials Needs more time for crosslinking Lack of control over the reaction kinetics of crosslinking Lower degrees of crosslinking
**Chemical cross-linking**	
+ Forming very strong bonds	Cell toxicity remains to be tested Needs washing to remove the residual cross-linker More expensive than physical cross-linking

## Data Availability

The data presented in this study are available on request from the corresponding author.

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
