# Peer review of "How to Improve Physico-Chemical Properties of Silk Fibroin Materials for Biomedical Applications?—Blending and Cross-Linking of Silk Fibroin—A Review"

_materials, 2021, doi:10.3390/ma14061510_

Round 1

Reviewer 1 Report

Article Review: How to improve physico-chemical properties of silk fibroin materials for biomedical applications? – blending and cross-linking of silk fibroin – a review

Journal:  Materials

Date: 1MAR21

Overview:

The authors provide a useful overview of silk fibroin materials and the approach to blend with other polymers and crosslink for improved physico-chemical properties. The manuscript is generally well-written with a logical discussion, though some minor grammar/wording issues should be resolved. There are a few instances where more context would enhance the discussion The following comments are offered to strengthen the quality of the final manuscript.

General Comments:

Lines 54: Change “biomaterials” to “biomaterial”

Line 88: Awkward phrasing – recommend rewording.

Line 91: “wieldy” should be “widely”?

Line 114: Change “have” to “has”

Line 125: Recommend changing wording to "This procedure must be" or something similar

Line 129: Should “1:8:2,5” have a decimal point rather than a comma?

Line 130: Recommend completing the sentence with “and ionic liquids”

Line 146-147: This sentence’s meaning is unclear.

Line 152: Change “using” to “used” and “exploiting” to “exploited”

Figure 3: The top circle text of “Bone Tissue Engineering”  is overlapped by adjacent circles. Recommend using “bring forward” if prepared in Power Point.

Line 167: Change “have” to “has”

Line 169: Consider including silk composite fibers here and/or in Table 8: Mitropoulos, A.N., et al. Noble metal composite porous silk fibroin aerogel fibers.  Materials, 2019, 12(6) 894.

Figure 4: Is there a higher resolution image available?

Figure 5: Is there a citation reference for this image or is this a new figure? Should explain "ALPHA 1–2 LDplus, 208 CHRIST"

Line 216: Change “to” to “for”

Figure 6: Image scale bars are unreadable

Figure 7: Is there a citation for this figure or is this new data/figure?

Line 270: Recommend changing “obtainment” to “obtained”

Table 3:  Is this new data or should there be a citation?

Line 325: A general audience may not understand what double staining is.

Line 350: Recommend deleting “were”

Line 350-351: Recommend "suggested further studies"

Line 360-361: Not sure what “achieves homeostasis” means in this context.

Line 381-383: It may be helpful to cite ranges if tensile strengths and Young's moduli

Line 403:  Define “hMSCs” acronym

Line 468: Recommend rewording “If it is about”

Line 470: Consider adding some context as to why curcumin used.

Line 489-490: This statement needs multiple citations or refer to Table 6 below.

Line 597: Recommend changing “Inaccuracies and discrepancies” to “Variabality”

Author Response

Manuscript ID: materials-1117293

Title: How to improve physico-chemical properties of silk fibroin materials for biomedical applications? – blending and cross-linking of silk fibroin – a review

Journal: Materials

Dear Reviewer, 

We would like to thank for the comments on our manuscript submitted to Materials for review process. We would like to thank also the Editor that gave us a chance to correct our manuscript. All changes made in our manuscript have been written in red. Please find below our answer to the comments.

Reviewer 1:

Lines 54: Change “biomaterials” to “biomaterial”

Dear Reviewer, thank you for your comment. We are sorry for that mistake. We removed “s” – it was a typo.

Line 88: Awkward phrasing – recommend rewording.

Dear Reviewer, thank you for your comment. The sentence has been redrafted: “Silk fibroin is built of repetitive protein sequences, what result in providing structural roles in arachnids and crustaceans life: web formation, egg protection, cocoon formation, building nest, and traps.”

Line 91: “wieldy” should be “widely”?

Dear Reviewer, thank you for your comment. It has been changed. We read the text carefully and corrected the remaining typos.

Line 114: Change “have” to “has”

Dear Reviewer, thank you for your comment. It has been changed. We read the text carefully and corrected the remaining typos.

Line 125: Recommend changing wording to "This procedure must be" or something similar

Dear Reviewer, thank you for your comment. The sentence has been redrafted.

Line 129: Should “1:8:2,5” have a decimal point rather than a comma?

Dear Reviewer, thank you for your comment. Yes, the point should be here. We are sorry for that mistake.

Line 130: Recommend completing the sentence with “and ionic liquids”

Dear Reviewer, thank you for your comment. It has been changed.

Line 146-147: This sentence’s meaning is unclear.

Dear Reviewer, thank you for your comment. The sentence has been redrafted.

Line 152: Change “using” to “used” and “exploiting” to “exploited”

Dear Reviewer, thank you for your comment. It has been changed.

Figure 3: The top circle text of “Bone Tissue Engineering”  is overlapped by adjacent circles. Recommend using “bring forward” if prepared in Power Point.

Dear Reviewer, thank you for your comment. It is impossible to do each circle forward, every time one is overlapped, one is forward. We changed places of applications, now it looks better.

Line 167: Change “have” to “has”

Dear Reviewer, thank you for your comment. It has been changed. We read the text carefully and corrected the remaining typos.

Line 169: Consider including silk composite fibers here and/or in Table 8: Mitropoulos, A.N., et al. Noble metal composite porous silk fibroin aerogel fibers.  Materials, 2019, 12(6) 894.

Dear Reviewer, thank you for your comment and proposition. We decided to add some information from this article. You can find this sentence in our article before Table 8 and in Table 8:

“Silk fibroin-based materials with addition of noble metals (tetrachloropalladate (II) (Na2PdCl4), potassium tetrachloroplatinate (II) (K2PtCl4), gold chloride trihydrate (HAuCl4·3H2O)) also can be found in the literature [181]. They can be used for biological sensing and energy storage applications [181].”

Figure 4: Is there a higher resolution image available?

Dear Reviewer, thank you for your comment and proposition. The higher resolution image is now available in manuscript.

Figure 5: Is there a citation reference for this image or is this a new figure? Should explain "ALPHA 1–2 LDplus, 208 CHRIST"

Dear Reviewer, thank you for your comment. It is a new figure. It is the name and company of equipment used to lyophilization process.

Line 216: Change “to” to “for”

Dear Reviewer, thank you for your comment. It has been changed. We read the text carefully and corrected the remaining typos.

Figure 6: Image scale bars are unreadable

Dear Reviewer, thank you for your comment. It has been changed.

Figure 7: Is there a citation for this figure or is this new data/figure?

Dear Reviewer, thank you for your comment. It is a new figure.

Line 270: Recommend changing “obtainment” to “obtained”

Dear Reviewer, thank you for your comment. It has been changed.

Table 3:  Is this new data or should there be a citation?

Dear Reviewer, thank you for your comment. These are the results from our previous report. There should be a citation here. We apologize for forgetting this.

Line 325: A general audience may not understand what double staining is.

Dear Reviewer, thank you for your comment. The information about method was added (simultaneous fluorescence staining of viable and dead cells).

Line 350: Recommend deleting “were”

Dear Reviewer, thank you for your comment. It has been changed.

Line 350-351: Recommend "suggested further studies"

Dear Reviewer, thank you for your comment. It has been changed.

Line 360-361: Not sure what “achieves homeostasis” means in this context.

Dear Reviewer, thank you for your comment. We decided to remove this fragment.

Line 381-383: It may be helpful to cite ranges if tensile strengths and Young's moduli

Dear Reviewer, thank you for your comment. We completed the missing information.

Line 403:  Define “hMSCs” acronym

Dear Reviewer, thank you for your comment. We completed the missing information. hMSCs - human bone marrow mesenchymal stem cells

Line 468: Recommend rewording “If it is about”

Dear Reviewer, thank you for your comment. We have replaced this snippet.

Line 470: Consider adding some context as to why curcumin used.

Dear Reviewer, thank you for your comment. We completed the missing information. Curcumin was used as antioxidant with anticancer activity.

Line 489-490: This statement needs multiple citations or refer to Table 6 below.

Dear Reviewer, thank you for your comment. We completed the missing references.

Line 597: Recommend changing “Inaccuracies and discrepancies” to “Variabality”

Dear Reviewer, thank you for your comment. We have replaced this snippet.

Reviewer 2 Report

The manuscript submitted to Materials entitled "How to improve physico-chemical properties of silk fibroin materials for biomedical applications? – blending and cross-linking of silk fibroin – a review" by Zielińska et al. presents a review about a subject that recently has attracted enormous attention. In general,  this revision is exquisitely descriptive and, at some point, confusing in some sections. The introduction is not clear, and it lacks focus. The authors refer to bone tissue engineering, but they do not give much information about this. Furthermore, in section 4, Figure 3 shows a lot more applications related to this polymer than bone tissue engineering. The introduction should be reorganized and direct the reader towards the importance of the polymer itself in general biomedical applications. Besides, a few aspects should be improved and clarified.

For instance:

1) A table of contents and an abbreviation table should be placed in the manuscript. Some of the abbreviations do not have the full term in parentheses.

2) Correct the document for typos, misspellings and wording. For instance:
Line 91: "wieldy"

3) in line 34, it is stated that reconstructive medicine develops at an alarming rate. Alarming typically has a negative connotation. Is this what the authors want to transmit?

4) From line 34 to line 41, there is not a single reference. Please refer to the work from where this information is based. In addition, in line 95, the text is highly fragmented and difficult to understand.
In the following lines, the text is confusing and need clarification, lines: 34 to 41; line 350.

5) The authors should add a section with the significant advances and those who might follow in the future on Silk Fibroin. Surely, after compiling all this information, the authors have found tendencies and how these materials can improve. This would increase the value of this review. 

Thank you

Author Response

Manuscript ID: materials-1117293

Title: How to improve physico-chemical properties of silk fibroin materials for biomedical applications? – blending and cross-linking of silk fibroin – a review

Journal: Materials

Dear Reviewer, 

We would like to thank for the comments on our manuscript submitted to Materials for review process. We would like to thank also the Editor that gave us a chance to correct our manuscript. All changes made in our manuscript have been written in red. Please find below our answer to the comments.

Reviewer 1:

The introduction is not clear, and it lacks focus. The authors refer to bone tissue engineering, but they do not give much information about this. Furthermore, in section 4, Figure 3 shows a lot more applications related to this polymer than bone tissue engineering. The introduction should be reorganized and direct the reader towards the importance of the polymer itself in general biomedical applications.

Dear Reviewer, thank you for your comment. New information was added in the beginning part of the manuscript.

Biopolymeric materials are commonly used in biomedical field, e. g.: wound healing, gene therapy, drug delivery, bone tissue engineering, cartilage, nerve and eye re-generation [1–6]. Increasing interest in new materials with potential use in biomedical field has been observed during the last few years [1–5,7]. In this manuscript, we re-viewed different usages of silk fibroin-based materials blended with other biopolymers and cross-linked with chemical agents, in biomedical applications based on previous researches. But one area of biomedical applications, deserves a greater introduction - bone tissue engineering.

  1. Kashirina, A.; Yao, Y.; Liu, Y.; Leng, J. Biopolymers as Bone Substitutes: A Review. Biomater Sci 2019, 7, 3961–3983, doi:10.1039/C9BM00664H.
  2. Aderibigbe, B.A.; Buyana, B. Alginate in Wound Dressings. Pharmaceutics 2018, 10, doi:10.3390/pharmaceutics10020042.
  3. Aguilar, A.; Zein, N.; Harmouch, E.; Hafdi, B.; Bornert, F.; Offner, D.; Clauss, F.; Fioretti, F.; Huck, O.; Benkirane-Jessel, N.; et al. Application of Chitosan in Bone and Dental Engineering. Molecules 2019, 24, doi:10.3390/molecules24163009.
  4. Zhang, J.; Xia, W.; Liu, P.; Cheng, Q.; Tahi, T.; Gu, W.; Li, B. Chitosan Modification and Pharmaceutical/Biomedical Applications. Marine Drugs 2010, 8, doi:10.3390/md8071962.
  5. Pollini, M.; Paladini, F. Bioinspired Materials for Wound Healing Application: The Potential of Silk Fibroin. Materials 2020, 13, doi:10.3390/ma13153361.
  6. Nosrati, H.; Pourmotabed, S.; Sharifi, E. A Review on Some Natural Biopolymers and Their Applications in Angiogenesis and Tissue Engineering. Appl Biotechnol Rep 2018, 5, 81–91, doi:10.29252/JABR.05.03.01.
  7. Sionkowska, A. Current Research on the Blends of Natural and Synthetic Polymers as New Biomaterials: Review. Prog Polym Sci 2011, 36, 1254–1276, doi:10.1016/j.progpolymsci.2011.05.003.

Besides, a few aspects should be improved and clarified.

For instance:

1) A table of contents and an abbreviation table should be placed in the manuscript. Some of the abbreviations do not have the full term in parentheses.

Dear Reviewer, thank you for your comment. The table of content and abbreviation list were added to manuscript.

2) Correct the document for typos, misspellings and wording. For instance:

Line 91: "wieldy"

Dear Reviewer, thank you for your comment. We are sorry for that mistake. We read the text carefully and corrected the remaining typos.

3) in line 34, it is stated that reconstructive medicine develops at an alarming rate. Alarming typically has a negative connotation. Is this what the authors want to transmit?

Dear Reviewer, thank you for your comment. The word “alarming” have been replaced to “increasing”.

4) From line 34 to line 41, there is not a single reference. Please refer to the work from where this information is based. In addition, in line 95, the text is highly fragmented and difficult to understand.

In the following lines, the text is confusing and need clarification, lines: 34 to 41; line 350.

Dear Reviewer, thank you for your comment. We added appropriate references in manuscript. Additionally we corrected other fragments of the text you mention.

5) The authors should add a section with the significant advances and those who might follow in the future on Silk Fibroin. Surely, after compiling all this information, the authors have found tendencies and how these materials can improve. This would increase the value of this review.

Dear Reviewer, thank you for your comment. We added some information to conclusion section. Additionally we are thinking that readers can find summary and some feature perspectives in this section.

This review can be used by wide group of scientists and researchers, who are working to obtain appropriate material useful in broadly understood biomedicine. The methods of physico-chemical and biological properties silk-based materials improvement were reported and discussed. Blending with other polymers and cross-linking can be used to improve silk fibroin-based materials properties, which can be desirable in biomedical field, especially in tissue engineering. Blending includes natural and synthetic polymers, cross-linking includes enzymatic, physical, and chemical processes. There are many articles regarding silk fibroin and complexes based on silk fibroin modified by the addition of second and third polymer or cross-linking process and it is hard to compare results reported by different research groups. The published results are rather consistent in the presented papers. Variability may result from the fact, that working with polymers obtained from natural sources is not easy. The physico-chemical properties of obtained silk fibroin are a little bit different in each batch of cocoons or source material. In addition, as for the results of mixtures of silk fibroin with polysaccharides, are probably caused by differences in the polymer molecular weight and deacetylation degree.

Some features make silk fibroin a promising base to obtain biomaterials for many clinical functions: the unique structure, biocompatibility, versatility in processing, availability of different morphologies (fibres, films, 3D porous structures, particles, hydrogels), options for genetic engineering of variations of silks, thermal stability, the ease of sterilization, surface chemistry for facile chemical modifications, and controllable degradation. Each of the silk fibroin-based systems has shown promising features for different biomedical applications. More research will have to be done before silk fibroin can be used for clinical trials and commercialized for tissue engineering applications, especially for bone tissue engineering because there is not too many in vivo studies with silk fibroin materials and these studies were done mostly in small animals, e.g. rats that do not sufficiency predict their performance in humans. A better understanding is needed regarding silk fibroin systems to create tissues which will be able to remodel similar to bone. Also, the long-term biocompatibility, biodegradability, and degraded products, along with the ability to tune silk morphologies for tissue-specific requirements are need to be better understood. It can be expected that with novel processing techniques, new silk fibroin-based composites will be developed in the future and open up even more possibilities for tissue engineering applications. The future for silk fibroin biomaterials to impact clinical needs appears promising and has big potential to bring viable strategies and innovations.

Reviewer 3 Report

The review “How to improve physico-chemical properties of silk fibroin materials for biomedical applications? – blending and cross-linking of silk fibroin – a review” is an overview on the possible application of silk fibroin in the biomedical field, describing the methods to improve its properties. In particular, the blending with other polymers and cross-linking of silk fibroin-based materials are reported.

This review is well organized and the study of the literature has been accurately performed. Therefore, the publication of this work is recommended; but after some revisions.

In particular:

- Introduction. Some works related to the production of natural aerogels by alternative techniques, like supercritical drying, should be added to complete the investigation of the literature. For this purpose, see, for instance, the works of: Smirnova and Gurikov, Aerogel production: Current status, research directions, and future opportunities, The Journal of Supercritical Fluids, 134, 2018, pp. 228-233; Baldino et al., A new tool to produce alginate-based aerogels for medical applications, by supercritical gel drying, Journal of Supercritical Fluids, 2019, 146, pp. 152-158; Li et al., Fabrication, characterization, and in vitro evaluation of biomimetic silk fibroin porous scaffolds via supercritical CO2 technology, Journal of Supercritical Fluids, 2019, 150, pp. 86-93; etc..

- From a chemical point of view, separation phenomena should be better explained to highlight the possible difficulty in blending together biopolymers, as well as the problems related to the crosslinking agents residue after processing that can be cytotoxic for cells.

- English can be improved.

Author Response

Manuscript ID: materials-1117293

Title: How to improve physico-chemical properties of silk fibroin materials for biomedical applications? – blending and cross-linking of silk fibroin – a review

Journal: Materials

Dear Reviewer, 

We would like to thank for the comments on our manuscript submitted to Materials for review process. We would like to thank also the Editor that gave us a chance to correct our manuscript. All changes made in our manuscript have been written in red. Please find below our answer to the comments.

Reviewer 3:

Introduction. Some works related to the production of natural aerogels by alternative techniques, like supercritical drying, should be added to complete the investigation of the literature. For this purpose, see, for instance, the works of: Smirnova and Gurikov, Aerogel production: Current status, research directions, and future opportunities, The Journal of Supercritical Fluids, 134, 2018, pp. 228-233; Baldino et al., A new tool to produce alginate-based aerogels for medical applications, by supercritical gel drying, Journal of Supercritical Fluids, 2019, 146, pp. 152-158; Li et al., Fabrication, characterization, and in vitro evaluation of biomimetic silk fibroin porous scaffolds via supercritical CO2 technology, Journal of Supercritical Fluids, 2019, 150, pp. 86-93; etc.

Dear Reviewer, thank you for your comment and proposition. We decided to add subsection to chapter: “Types of silk fibroin based biomaterials”.

4.6. Aerogels

Aerogels are ultralight materials comprised of a microporous solid in which the dispersed phase is a gas [75]. They are characterized by fine, open-pore structure resulting in low densities (0.003–0.15 kg/m3), high porosity and large surface areas (500–1000 m2/g) [76]. The surface area, pore size, mechanical and physico-chemical properties of aerogels can be tailored [76]. Few drying technologies for aerogel production can be pointed: ambient drying (ambient pressure, room or slightly elevated temperature), freeze drying (vacuum with P<100 mbar; −70 < T<−20 °C), direct supercritical drying (high temperature), supercritical drying by CO2 extraction (T > 31 °C, P > 74 bar) [76–80]. Aerogels can find application in many fields, including packaging materials, cosmetics and medicine in particular drug delivery systems, tissue engineering and implants. There are only a few literature reports on silk fibroin-based aerogels. Li et al. [79] and Baldino et al. [81] reported silk fibroin-based aerogels and silk fibroin aerogels loaded with ascorbic acid, for nerve regeneration and nanomedicine applications, respectively. Following to Najberg et al. [82], aerogel sponges of silk fibroin, hyaluronic acid and heparin can be used for soft tissue engineering. Goimil et al. [83] prepared and re-ported silk fibroin aerogel/poly(ℇ-caprolactone) scaffolds containing dexamethasone for bone regeneration. According to Li et al. [84], silk fibroin/gelatin nanoparticles aerogel is a promising bone tissue engineering material.

  1. Alemán, J.V.; Chadwick, A.V.; He, J.; Hess, M.; Horie, K.; Jones, R.G.; Kratochvíl, P.; Meisel, I.; Mita, I.; Moad, G.; et al. Definitions of terms relating to the structure and processing of sols, gels, networks, and inorganic-organic hybrid materials (IUPAC Recommendations 2007). Pure Appl Chem 2007, 79, 1801–1829, doi:10.1351/pac200779101801.
  2. Smirnova, I.; Gurikov, P. Aerogel Production: Current Status, Research Directions, and Future Opportunities. J Supercrit Fluid 2018, 134, 228–233, doi:10.1016/j.supflu.2017.12.037.
  3. Baldino, L.; Cardea, S.; Scognamiglio, M.; Reverchon, E. A New Tool to Produce Alginate-Based Aerogels for Medical Applications, by Supercritical Gel Drying. J Supercrit Fluid 2019, 146, 152–158, doi:10.1016/j.supflu.2019.01.016.
  4. Baldino, L.; Cardea, S.; Reverchon, E. Natural Aerogels Production by Supercritical Gel Drying. Chem Engineer Trans 2015, 43, 739–744, doi:10.3303/CET1543124.
  5. Zh, L.; L, W.; Hl, D.; Xy, W.; Js, L.; Z, Z. Fabrication, characterization, and in vitro evaluation of biomimetic silk fibroin porous scaffolds via supercritical CO2 technology. J. Supercrit. Fluids 2019, 150, 86–93.
  6. Baldino, L.; Concilio, S.; Cardea, S.; Reverchon, E. Interpenetration of Natural Polymer Aerogels by Supercritical Drying. Polymers 2016, 8, doi:10.3390/polym8040106.
  7. Lucia Baldino; Stefano Cardea; Ernesto Reverchon Loaded Silk Fibroin Aerogel Production by Supercritical Gel Drying Process for Nanomedicine Applications. Chem Engineer Trans 2016, 49, 343–348, doi:10.3303/CET1649058.
  8. Najberg, M.; Haji Mansor, M.; Taillé, T.; Bouré, C.; Molina-Peña, R.; Boury, F.; Cenis, J.L.; Garcion, E.; Alvarez-Lorenzo, C. Aerogel Sponges of Silk Fibroin, Hyaluronic Acid and Heparin for Soft Tissue Engineering: Composition-Properties Relationship. Carbohydrate Polymers 2020, 237, 116107, doi:10.1016/j.carbpol.2020.116107.
  9. Goimil, L.; Santos-Rosales, V.; Delgado, A.; Évora, C.; Reyes, R.; Lozano-Pérez, A.A.; Aznar-Cervantes, S.D.; Cenis, J.L.; Gómez-Amoza, J.L.; Concheiro, A.; et al. ScCO2-Foamed Silk Fibroin Aerogel/Poly(ε-Caprolactone) Scaffolds Containing Dexamethasone for Bone Regeneration. J CO2 Util 2019, 31, 51–64, doi:https://doi.org/10.1016/j.jcou.2019.02.016.
  10. Li, D.; Chen, K.; Duan, L.; Fu, T.; Li, J.; Mu, Z.; Wang, S.; Zou, Q.; Chen, L.; Feng, Y.; et al. Strontium Ranelate Incorporated Enzyme-Cross-Linked Gelatin Nanoparticle/Silk Fibroin Aerogel for Osteogenesis in OVX-Induced Osteoporosis. ACS Biomater. Sci. Eng. 2019, 5, 1440–1451, doi:10.1021/acsbiomaterials.8b01298.

- From a chemical point of view, separation phenomena should be better explained to highlight the possible difficulty in blending together biopolymers, as well as the problems related to the crosslinking agents residue after processing that can be cytotoxic for cells.

Dear Reviewer, thank you for your comment and proposition. We decided to add some information about miscibility studies in Blending section.

In an aim to produce mixtures characterized by unique structural and mechanical properties, blending of biopolymers can be used. For the production of biomaterials with better properties, mixtures of two or more polymers can be used. The natural polymers, which are blend components, can be combined in the molten state (also known as melt mixing) [88] or they can be mixed as aqueous solutions in appropriate solvents [89–91]. If it is about melt mixing, the reaction between solids at high pressure and high temperature can be devastating to natural polymers, belonging to protein group. The result of these high parameters may be denaturation and degradation of natural polymers [11]. The blending of biopolymers as aqueous solutions in appropriate solvents can be a solution of the above problem. But some of the natural polymers are insoluble in common solvents. That’s why, miscibility studies can be desired and used. Four main groups of miscibility studies can be pointed out: methods based on the determination of optical homogeneity of the mixture; methods for the determination of glass transition temperatures; methods for the direct determination of interactions on molecular levels; indirect methods for the miscibility [11]. Fourier transform infrared spectroscopy (FTIR), viscometry and differential scanning calorimetry (DSC) are listed as the most common and easy techniques to investigate miscibility [11]. FTIR spectroscopy is used to study specific molecular bonding interactions in polymer blends. The changes in IR spectra (new bands, disappearance of some component bands, shifts in the specific bands) are characteristic for miscible systems. For immiscible blends, spectrum can reflect two individual components [11]. In viscometry method, experimental parameters of the mixture (b) and intrinsic viscosity (η) are comparing with their ideal (calculated) values, according to Krigbaum et al. [92] and Garcia et al. [93]. DSC is the most commonly used technique for determination of glass transition temperature (Tg). Determination of the number of glass transition temperature is the main method for investigation of the number of amorphous phases in polymer systems. Each Tg corresponds to one amorphous phase and DSC provides determination of the number of the phases that coexist in a polymer mixture [11].

  1. Sionkowska, A. Current Research on the Blends of Natural and Synthetic Polymers as New Biomaterials: Review. Prog Polym Sci 2011, 36, 1254–1276, doi:10.1016/j.progpolymsci.2011.05.003.
  2. Rogovina, S.Z.; Vikhoreva, G.A. Polysaccharide-Based Polymer Blends: Methods of Their Production. Glycoconj J 2006, 23, 611, doi:10.1007/s10719-006-8768-7.
  3. Ghaeli, I.; de Moraes, M.; Beppu, M.; Lewandowska, K.; Sionkowska, A.; Ferreira-da-Silva, F.; Ferraz, M.; Monteiro, F. Phase Behaviour and Miscibility Studies of Collagen/Silk Fibroin Macromolecular System in Dilute Solutions and Solid State. Molecules 2017, 22, 1368, doi:10.3390/molecules22081368.
  4. Lewandowska, K.; Sionkowska, A.; Grabska, S.; Kaczmarek, B.; Michalska, M. The Miscibility of Collagen/Hyaluronic Acid/Chitosan Blends Investigated in Dilute Solutions and Solids. J Mol Liq 2016, 220, 726–730, doi:10.1016/j.molliq.2016.05.009.
  5. Sionkowska, A.; Lewandowska, K.; Płanecka, A. Miscibility and Physical Properties of Chitosan and Silk Fibroin Mixtures. Journal of Molecular Liquids 2014, 198, 354–357, doi:10.1016/j.molliq.2014.07.033.
  6. Krigbaum, W.R.; Wall, F.T. Viscosities of Binary Polymeric Mixtures. J Polym Sci 1950, 5, 505–514, doi:10.1002/pol.1950.120050408.
  7. Garcı́a, R.; Melad, O.; Gómez, C.M.; Figueruelo, J.E.; Campos, A. Viscometric Study on the Compatibility of Polymer–Polymer Mixtures in Solution. European Polymer Journal 1999, 35, 47–55, doi:10.1016/S0014-3057(98)00106-2.

We also added information about toxicity effect of cross-linking.

The chemical-cross-linking method is considered to be the most effective and predictable [198]. It provides for formation of  very strong bonds. The disadvantages of chemical cross-linking is  high price (chemical cross-linking is more expensive than physical cross-linking), necessity to toxicity testing and necessity to remove the residual of cross-linker [202]. Some detoxifying strategies have been proposed to exclude toxic effect of cross-linker residual. For instance, if the cross-linker has aldehyde groups (e.g. glutaraldehyde), washing of cross-linked scaffolds with solutions containing free amino groups or amino acid solutions (e.g. glycine) can be used and it led to the removal free aldehyde groups [200,202]. If it is about carbodiimide agents (e.g. N-(3-dimethylaminopropyl)-N’-ethylcarbodiimide hydrochloride – EDC), all of the residues are water soluble and they can be washed out of the cross-linked scaffold construct easily by distilled water after the completion of the cross-linking reaction [200]. The concentration of cross-linking agent also have big influence on toxicity effect of cross-linking process [206]. For example, concentration up to 8% of glutaraldehyde have shown no toxicity for cross-linking [202]. According to literature, fabrication process is also very important [207]. For instance, glutaraldehyde can be added to collagen/chitosan material before freeze-drying procedure to obtain better properties after cross-linking process, however the toxicity effect of glutaraldehyde in this manner can be higher than during the protocol when glutaraldehyde was added after freeze-drying stage [202,207]. To sum up, the type and amount of material, concentration of cross-linker and fabrication protocol affect the biocompatibility of material [200,202].

  1. Oryan, A.; Kamali, A.; Moshiri, A.; Baharvand, H.; Daemi, H. Chemical Crosslinking of Biopolymeric Scaffolds: Current Knowledge and Future Directions of Crosslinked Engineered Bone Scaffolds. Int J Biol Macromol 2018, 107, 678–688, doi:10.1016/j.ijbiomac.2017.08.184.
  2. Chirila, T.V.; Suzuki, S.; Papolla, C. A Comparative Investigation of Bombyx Mori Silk Fibroin Hydrogels Generated by Chemical and Enzymatic Cross-Linking: Bombyx Mori Silk Fibroin Hydrogels. Biotechnology and Applied Biochemistry 2017, 64, 771–781, doi:10.1002/bab.1552.
  3. Ostrowska-Czubenko, J.; Pieróg, M.; Gierszewska, M. Modification of chitosan - a concise overview. Wiadomości Chemiczne 2016, 70, 9–10.
  4. Dmour, I.; Taha, M. Natural and semisynthetic polymers in pharmaceutical nanotechnology. In Organic Materials as Smart Nanocarriers for Drug Delivery; 2018 ISBN 978-0-12-813663-8.
  5. Ruijgrok, J.M.; De Wijn, J.R.; Boon, M.E. Optimizing Glutaraldehyde Crosslinking of Collagen: Effects of Time, Temperature and Concentration as Measured by Shrinkage Temperature. J Mater Sci: Mater Med 1994, 5, 80–87, doi:10.1007/BF00121695.
  6. Delgado, L.M.; Bayon, Y.; Pandit, A.; Zeugolis, D.I. To Cross-Link or Not to Cross-Link? Cross-Linking Associated Foreign Body Response of Collagen-Based Devices. Tissue Eng Part B Rev 2015, 21, 298–313, doi:10.1089/ten.teb.2014.0290.
  7. Beauchamp, R.O.; St Clair, M.B.; Fennell, T.R.; Clarke, D.O.; Morgan, K.T.; Kari, F.W. A Critical Review of the Toxicology of Glutaraldehyde. Crit Rev Toxicol 1992, 22, 143–174, doi:10.3109/10408449209145322.

- English can be improved.

Dear Reviewer, thank you for your comment. We did our best to improve English.

Round 2

Reviewer 2 Report

In general, I am pleased with the author's responses to my concerns and the general improvements made to the manuscript after the initial assessment.

Thank you

Reviewer 3 Report

The authors answered and performed all the modifications proposed by the Reviewer and improved the manuscript. The publication is recommended.